



# ₁ Rapid attribution of the August 2016 flood-inducing extreme ₂ precipitation in south Louisiana to climate change

₃ Karin van der Wiel[1,2], Sarah B. Kapnick[2], Geert Jan van Oldenborgh[3], Kirien Whan[3], Sjoukje Philip[3],
₄ Gabriel A. Vecchi[2], Roop K. Singh[4], Julie Arrighi[4], Heidi Cullen[5]
₅ [1]Program in Atmospheric and Oceanic Sciences, Princeton University, Princeton, U.S.
₆ [2]Geophysical Fluid Dynamics Laboratory (GFDL), National Oceanic and Atmospheric Administration, Princeton, U.S.
₇ [3]Royal Netherlands Meteorological Institute (KNMI), De Bilt, Netherlands
₈ [4]Red Cross Red Crescent Climate Centre, The Hague, Netherlands
₉ [5]Climate Central, Princeton, U.S.

₁₀ *Correspondence to*: Karin van der Wiel (kwiel@princeton.edu) or Geert Jan van Oldenborgh (oldenborgh@knmi.nl).

₁₁ **Abstract.**

₁₂ A stationary low pressure system and elevated levels of precipitable water provided a nearly continuous source of

₁₃ precipitation over Louisiana, United States (U.S.) starting around 10 August, 2016. Precipitation was heaviest in the region

₁₄ broadly encompassing the city of Baton Rouge, with a three-day maximum found at a station in Livingston, LA (east of

₁₅ Baton Rouge) from 12–14 August, 2016 (648.3 mm, 25.5 inches). The intense precipitation was followed by inland flash

₁₆ flooding and river flooding and in subsequent days produced additional backwater flooding. On 16 August, Louisiana

₁₇ officials reported that 30,000 people had been rescued, nearly 10,600 people had slept in shelters on the night of 14 August,

₁₈ and at least 60,600 homes had been impacted to varying degrees. As of 17 August, the floods were reported to have killed at

₁₉ least thirteen people. As the disaster was unfolding, the Red Cross called the flooding the worst natural disaster in the U.S.

₂₀ since Super Storm Sandy made landfall in New Jersey on 24 October, 2012. Before the floodwaters had receded, the media

₂₁ began questioning whether this extreme event was caused by anthropogenic climate change. To provide the necessary

₂₂ analysis to understand the potential role of anthropogenic climate change, a rapid attribution analysis was launched in real-

₂₃ time using the best readily available observational data and high-resolution global climate model simulations.

₂₄       The objective of this study is to show the possibility of performing rapid attribution studies when both observational

₂₅ and model data, and analysis methods are readily available upon the start. It is the authors aspiration that the results be used

₂₆ to guide further studies of the devastating precipitation and flooding event. Here we present a first estimate of how

₂₇ anthropogenic climate change has affected the likelihood of a comparable extreme precipitation event in the Central U.S.

₂₈ Gulf Coast. While the flooding event of interest triggering this study occurred in south Louisiana, for the purposes of our

₂₉ analysis, we have defined an extreme precipitation event by taking the spatial maximum of annual 3-day inland maximum

₃₀ precipitation over the region: 29–31 ºN, 85–95 ºW, which we refer to as the Central U.S. Gulf Coast. Using observational

₃₁ data, we find that the observed local return time of the 12-14 August precipitation event in 2016 is about 550 years (95%



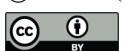

confidence interval (C.I.): 450-1450). The probability for an event like this to happen anywhere in the region is presently 1
in 30 years (C.I. 11-110). We estimate that these probabilities and the intensity of extreme precipitation events of this return
time have increased since 1900. A Central U.S. Gulf Coast extreme precipitation event has effectively become more likely in
2016 than it was in 1900. The global climate models tell a similar story, with the regional probability of 3-day extreme
precipitation increasing due to anthropogenic climate change by more than a factor 1.4 in the most accurate analyses. The
magnitude of the shift in probabilities is greater in the 25 km (higher resolution) climate model than in the 50 km model. The
evidence for a relation to El Niño half a year earlier is equivocal, with some analyses showing a positive connection and
others none.

## 1 Introduction

In the second week of August, a storm system developed in the United States (U.S.) Gulf Coast region and resulted in
intense precipitation across south Louisiana in the region surrounding the city of Baton Rouge. The highest concentration of
precipitation fell over the 3-day period of 12-14 August (Figure 1a-d). Saturday, 13 August experienced the greatest total
magnitude of precipitation and the broadest surface area of intense precipitation during the storm. The National Oceanic and
Atmospheric Administration (NOAA) Climate Prediction Center (CPC) unified gauge-based gridded analysis of daily
precipitation exhibits 25×25 km area boxes with precipitation maxima reaching up to 534.7 mm (21.1 inches) over the 3-day
period. In station observations (a single point), a rain gauge in Livingston, LA (east of Baton Rouge) experienced an even
higher 3-day precipitation total of 648.3 mm (25.5 inches). In places, the 3-day precipitation totals in Louisiana exceeded
three times that of the climatological August totals (historical average total precipitation that occurs over 31-days, Figure 1e)
and three times the average annual 3-day precipitation maximum (Figure 1f).
The intense precipitation formed due to a low pressure system that originated near Florida/Alabama on 5 August. At
that time the National Hurricane Center stated that it might transform into a tropical depression after moving to the Gulf of
Mexico (Schleifstein 2016). Instead the system remained over land and moved westward slowly. On 12 August it became
near-stationary over Louisiana (Figure 1a-c) allowing for the continuous development of thunderstorms in localized area to
the south and southeast of the low pressure center. The stationary storm system and anomalously moist atmospheric
conditions (precipitable water exceeding 65 mm) created optimal conditions for high precipitation efficiencies and intense
precipitation rates. Though the system had a warm-core and some similarities to a tropical depression, it never formed the
closed surface wind circulation about a well-defined center that are needed to be classified as one (National Weather Service

2016).





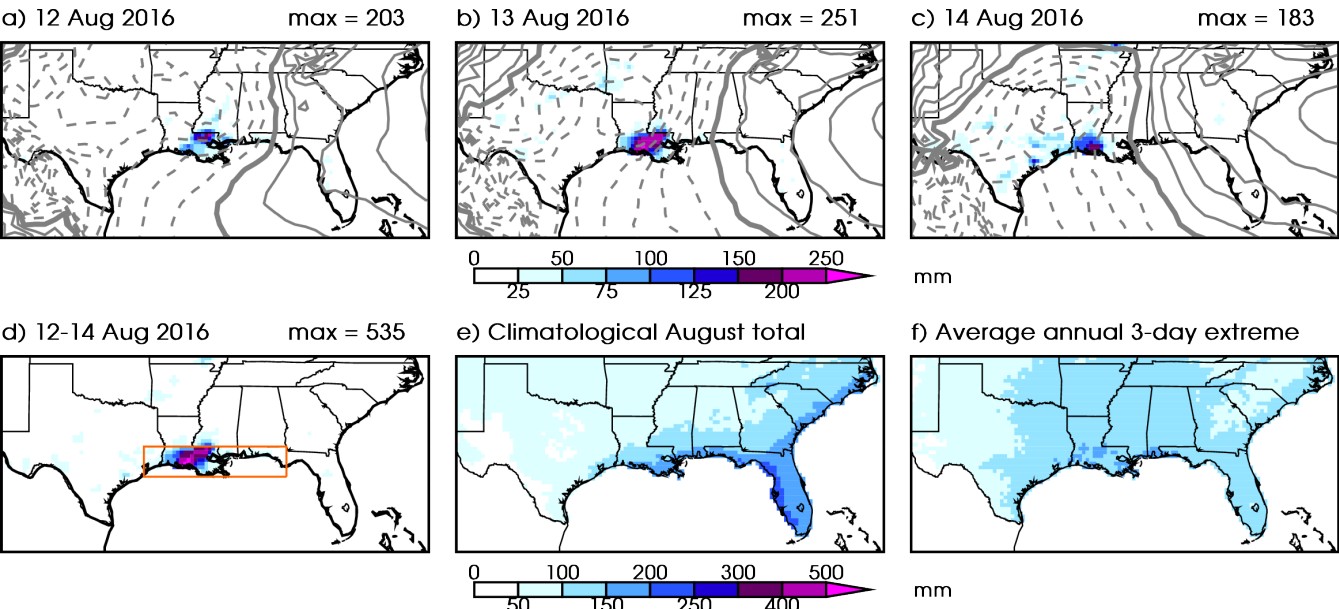

**Figure 1**: (a,b,c) Daily precipitation (shaded colors) and sea level pressure (grey contours, interval 1 hPa, 1015 hPa contour thickened, lower contours dashed) for 12, 13 and 14 August, 2016. (d) 3-day precipitation sum 12-14 August, 2016. (e) August climatological total precipitation (1948-2015). (f) Average annual maximum 3-day precipitation event (1948-2015). Orange box in (d) shows the geographic region used for the analysis (29º-31ºN, 85º-95ºW). Data from CPC unified gauge-based analysis of daily precipitation over the contiguous U.S. (2016 data from the real time archive) and ECMWF operational analysis.

Historic freshwater flooding in the region encompassing Baton Rouge, Louisiana followed the extreme precipitation event. Provisional reports from 18 August, 2016 showed streamgauges managed by the United States Geological Survey (USGS) registering above flood stage levels at 30 sites and found that out of 261 sites in all of Louisiana 50 were overtopped by floodwaters (Burton and Demas 2016). This was a complex event where provisional data from the USGS showed rivers responding to local precipitation as well as upstream and downstream conditions (Figure 2). For example, on the Comite River, a major drainage river for North Baton Rouge and its outlying districts, the provisional gauge height data exceeded the National Weather Service (NWS) flood stage from 12-16 August and even exceeded the previous height record (set 19 May, 1953). The Comite River hit its NWS flood stage level before the maximum precipitation fell in Central U.S. Gulf Coast (Figure 1d). Floodwaters were slow to recede due to flood stages downstream causing backwater flooding (upstream flooding caused by conditions downstream) in many neighborhoods (Burton and Demas 2016). Further downstream on the Amite River, provisional data showed that water levels exceeded the NWS floodstage from 13-23 August and also exceeded the previous height record (set 25 April, 1977). Its levels declined more slowly and did not fall below floodstage until late on 23 August, due to drainage from the Comite and other tributaries upstream that hit peak floodstage days earlier (Burton and Demas 2016).





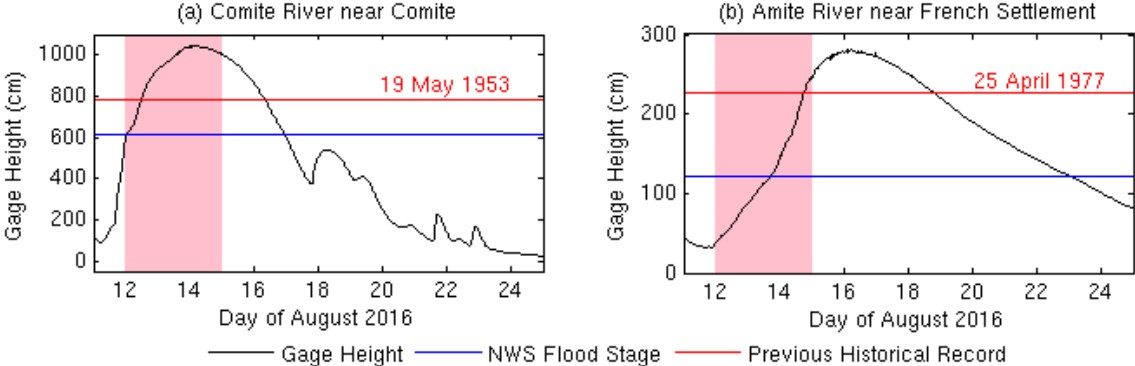

**Figure 2**: Hydrographs of gauge levels, NWS flood stage value and previous historical record for USGS station (a) 07378000 on the Comite River and (b) 07380200 on the Amite River. Observed streamgauge information downloaded 25 August, 2016 from the USGS: <http://waterdata.usgs.gov/la/nwis/uv?>; provisional USGS data is subject to adjustment: http://help.waterdata.usgs.gov/policies/provisional-data-statement.

On 12 August the NWS issued flash flood warnings for parishes in south Louisiana, and activated the national Emergency Alert System which urged residents to move to higher ground. The Louisiana Coast Guard, National Guard, and civilian volunteers mobilized to rescue over 30,000 people from their flooded homes and cars (Broach 2016). By August 14, the federal government declared a major disaster, indicating that the severity of damage exceeded the local and state governments' combined capability to respond, initiating federal assistance for individuals and public infrastructure (Davies 2016, FEMA 2016, Stafford Disaster Relief and Emergency Assistance Act). The flooding impacted the state's agriculture industry with losses estimated in excess of $110 million (Allen and Burgess 2016). Initial estimates also show that at least 60,600 homes were damaged, and thirteen people were killed due to the floods (Strum 2016). The American Red Cross, with FEMA and other federal and local agencies, provided shelter and emergency relief for 10,600 people initially displaced by the disaster, and the American Red Cross estimates that its ongoing relief efforts will cost $30 million (American Red Cross 2016). To date, more than 110,000 people have registered for federal disaster assistance (FEMA, 2016). FEMA has made grants totalling $107 million available to disaster survivors for serious needs including temporary rental assistance, and $20 million in advance payments for National Flood Insurance policyholders (FEMA 2016).

South Louisiana is a region where a number of phenomena can lead to flooding. For example, as a coastal region, it can experience saltwater flooding from a storm surge, when the low pressure and winds of a storm moving towards the coastline push coastal saltwater inland. This occurred in August 2005 when Hurricane Katrina impacted a broad swath of the Gulf Coast, including New Orleans, LA, with a large storm surge. Inland, precipitation can directly cause pluvial flooding by producing runoff in a region independent of a body of water (i.e. when more rain falls than can be soaked up by the ground) or fluvial flooding when water levels exceed the capacity of the river environment. For inland freshwater flooding, land surface conditions prior to an extreme precipitation event may increase the susceptibility of a region to both types of flooding, by saturating the soil (Tramblay et al. 2010, De Michele and Salvadori 2002) or increasing river levels (Pinter





2006). Inland flood conditions can also be induced by water flowing through the river system after a storm due to capacity limitations, as evident along the Amite River in August 2016 (Figure 2b) due to upstream flood conditions making their way downstream. Flooding can be influenced by remote meteorological conditions as river networks connect regions over vast areas. Louisiana had most recently experienced widespread inland flooding in March-April 2016. Although inland freshwater flooding occurs due to a combination of the level of extreme precipitation and its interaction with the land surface and river system, including human modifications to those systems and responses to events, we have chosen to focus our rapid attribution study on one portion of the problem: understanding the present and potentially climate change-influenced probability of extreme precipitation events like the one which occurred in August 2016.

Synoptic forcing for precipitation extremes in the Gulf Coast region includes both mid-latitude weather (cold core systems fueled by baroclinic instability), and tropical weather (warm core systems with barotropic instability). Extreme precipitation has historically been classified into 3 types of events: frontal systems, tropical systems, and air mass events. Each of these categories can be further broken down; e.g. tropical systems ranging from easterly waves to hurricanes, frontal systems including interactions between the polar jet and moist air masses from the Gulf, squall lines, or mesoscale convective systems, and air mass systems that may include heavy rainfall from upper air disturbances, or convective storms that form because of daytime heating (Keim and Faiers 1996). The variety of weather systems that can give rise to precipitation extremes in the region complicates the statistical analysis of the extremes and requires climate models to capture the entire distribution in a realistic manner. Also, the response to radiative forcing may be non-linear: thermodynamic and/or dynamic changes may be different for each weather type.

In this article, we analyze the historical context and changes in statistics of extreme precipitation like the one experienced during August 2016 in south Louisiana by defining an extreme event by its local or regional maximum 3-day precipitation. We have focused our analysis on stations or land surface grid cells in the region: 29–31 ºN, 85–95 ºW (illustrated by the red box in Figure 1d), which we hereafter refer to as the "Central U.S. Gulf Coast". Here we report the results of our rapid attribution study conducted by several organizations within two weeks of the event. The need for a rapid attribution study arises from the current intense public discussion that results from the significant societal impacts of this particular event and a continuous general interest in climate change. Media coverage following the event has linked into the growing body of scientific evidence that precipitation extremes are expected to increase due to the greater moisture content of a warmer atmosphere following Clausius-Clapeyron scaling (O'Gorman 2015, Lenderink and Attema 2015, Scherrer et al, 2016): e.g. "Disasters like Louisiana floods will worsen as planet warms, scientists warn" (Milman 2016), "Flooding in the South looks a lot like climate change" (Bromwich 2016). However, specific scientific statements for the event as observed in south Louisiana cannot be made based on general assessments of the connection of global warming and extreme rainfall. While attribution studies at a more traditional scientific pace (several months up to a year later) are important and add to scientific understanding of changing extremes, reporting results recently after an extreme event may enhance the societal



understanding of climate change and extreme weather, and provide often requested information for management decisions
following the event.

The methodologies employed in this study are used regularly in the literature and were previously applied to the
rapid attribution of the French and German 2016 flooding event (Van Oldenborgh et al. 2016) and of Storm Desmond over
the UK in 2015 (Van Oldenborgh et al. 2015). The presented analysis builds upon these methodologies for anthropogenic
climate change attribution and also explores the role of climate variability. The trends and internal climate variability of
extreme precipitation is investigated in station observations, gridded gauge-based precipitation analysis, and high-resolution
global climate model simulations. Since this paper aims to provide a first attribution assessment of the 2016 south Louisiana
extreme event, we have provided a detailed data and methods section (Section 2) in which our data sets, statistical
calculations for return periods and trends and data set validation methodologies are described. The rest of the paper is
organized as follows: Section 3 provides observational analysis. In Section 4 we evaluate the suitability of the global climate
models. Model analysis is provided in Section 5. Section 6 synthesizes our conclusions. In Section 7 we provide a detailed
discussion of crucial assumptions and their potential impact on the results, further avenues of research and implications of
this work.

## 2 Data and methods

### 2.1 Observational data

We utilize both point station observations and gridded analysis in this paper. The point station data are from the Global
Historical Climatology Network daily product (GHCN-D) version 3.22 (Menne et al. 2012, 2016). The data set provides
daily observations for stations worldwide. Data is quality controlled before becoming available in near-real time. Inside the
defined Central U.S. Gulf Coast (Figure 1d), 324 stations with a minimum of 10 years of data are available for the period
1891 to present (August 2016). However, not all stations provide data for the entire period, and spatial proximity between
stations means that not all data points provide independent information. Therefore for some analyses a smaller selection of
the available stations is taken into account. Selection criteria are described in the relevant sections.

The gridded analysis used here is the product of the NOAA Climate Prediction Center (CPC) Unified Gauge-Based
Analysis of Daily Precipitation over the contiguous U.S. (Higgins et al. 2000). The data set interpolates point station data on
a 0.25°×0.25° uniform latitude-longitude grid, based on the optimal interpolation scheme of Gandin and Hardin (1965). The
CPC dataset covers the period 1 January 1948 to present (August 2016), data from 2007 onwards has been made available in
real time. Because this is a gridded product, daily precipitation sums represent an areal average (0.25°×0.25°) rather than a
point measurement. Therefore precipitation extremes are expected to be of smaller magnitude in the gridded product (Chen
and Knutson 2008), as was noted for the south Louisiana event above (3-day total maxima of 534.7 mm in the CPC gridded
versus 648.3 mm in the point station data). The gridded analysis and the individual station data are not independent, as the





precipitation station data is the underlying source for the gridded analysis; consequently, changes in gauge station density in
space and time (as discussed above for GHCN-D) also impact the gridded analysis. We note that, for comparisons with
climate models - in which precipitation represents area averages, and not point values - the area-averaged precipitation
values from the gridded analysis are likely more meaningful for comparison with models than point station data (Chen and
Knutson 2008, Eggert et al, 2015).
We use the National Aeronautics and Space Administration (NASA) Goddard Institute for Space Science (GISS)
surface temperature analysis (GISTEMP, Hansen et al. 2010) for estimates of the development of global mean surface
temperature over time. This gridded data set is based on the GHCN point station data over land, NOAA Extended
Reconstructed Sea Surface Temperature (ERSST, Huang et al. 2015) version 4 over oceans and Scientific Committee on
Antarctic Research (SCAR) point station data for Antarctica.

## 2.2 Model and experiment descriptions

Many of the meteorological phenomena that cause extreme precipitation at the U.S. Gulf Coast are small-scale, therefore
only high-resolution models can simulate them realistically. We verified that the Royal Netherlands Meteorological Institute
(KNMI) EC-Earth 2.3 T159 experiments (~150km, Hazeleger et al. 2012)  and the United Kingdom (U.K.) Met Office
HadGEM3-A N216 (~60km, Christidis et al. 2013) models do not realistically represent precipitation extremes in the region.
We therefore use two higher-resolution global climate models in our analysis from the NOAA Geophysical Fluid
Dynamics Laboratory (GFDL). Both models were developed from the GFDL Coupled Model version 2.1 (CM2.1, Delworth
et al. 2006) using a cubed-sphere finite volume dynamical core (Putman and Lin 2007) with 32 vertical levels. Atmospheric
physics are taken from the GFDL Coupled Model version 2.5 (CM2.5, Delworth et al. 2006, 2012). The two models share
the same ocean and sea ice components with a 1º horizontal resolution, but differ in their atmosphere and land horizontal
resolution. In the Forecast-oriented Low Ocean Resolution model (FLOR, Vecchi et al. 2014), there are 180 points along
each cubed-sphere finite volume dynamical core face (FV3-C180), which relates to a resolution of 0.5º per cell along the
Equator. This has been interpolated to a 0.5º×0.5º uniform latitude-longitude grid. In the high-resolution version of the
model (HiFLOR, Murakami et al. 2016), there are 384 points along each face (FV3-C384) on the cubed-sphere finite volume
dynamic core, which relates to a resolution of 0.23º per cell along the Equator. This has been interpolated to a 0.25º×0.25º
uniform latitude-longitude grid. For FLOR we use a flux-adjusted version of the model (FLOR-FA), in which atmosphere-
to-ocean fluxes of momentum, enthalpy and freshwater are adjusted to bring the simulated fields closer to their observed
climatological state. The adjustment method is described in detail in Vecchi et al. (2014). Descriptions on how to access the
data used in this study are provided in the Data Availability section.
Table 1 describes six different global coupled model experiments that have been performed using FLOR-FA and
HiFLOR, which —for each model— differ in the type of radiative forcing that is prescribed, thus allowing us to assess the
impact of radiative forcing on the statistics of weather extremes in these models. With FLOR-FA there are two sets of





experiments. First, we made use of a multi-centennial integration in which values of radiative forcing agents (solar forcing,
anthropogenic and natural aerosols, well-mixed greenhouse gases, ozone, etc.) are prescribed to remain at levels
representative of a particular time - the mid-19[th] century in this case (Jia et al. 2016); radiative forcing agents are prescribed
at the 1860 values following the protocol of the Fifth Coupled Model Intercomparison Project (CMIP5, Taylor et al. 2009).
These types of experiments with global climate models are often referred to as "control" experiments ("pre-industrial
control" in this particular case) but here we label this class of experiments as "static radiative forcing" experiments, since
with HiFLOR we fix radiative forcing at a number of levels. In the static radiative forcing experiments the years of the
integration bear no relation to the real world calendar. The second set of experiments with FLOR-FA is a suite of five
realizations (or "ensemble members") in which the radiative forcing is prescribed to follow estimates of past and future
radiative forcing changes over the period 1861-2100 (Jia et al. 2016); the forcing agents for the period 1861-2005 are
prescribed to follow the CMIP5 historical experiment protocol, and for the period 2005-2100 they follow the CMIP5
Representative Concentration Pathway 4.5 (RCP4.5), which represents the medium range greenhouse gas emissions scenario
(Van Vuuren et al. 2011).  The five realizations of 1861-2100 experiments differ only in their initial conditions on January 1,
1861, which are taken from five different years from the long FLOR-FA preindustrial static forcing experiment. In these
experiments, the calendar of the experiments is connected to the history of radiative forcing - but the internal climate
variations (e.g., El Niño events) and weather fluctuations (e.g., individual storms) are not constrained to follow their
observed sequence. The static climate experiment has a slow drift because the slow climate components, notably the deep
ocean, were not in equilibrium at the beginning of the run, this is most noticeable in the first 1000 years of the integration.

**Table 1:** Global coupled model experiments performed with the FLOR-FA and HiFLOR models.

| Model | Type of forcing | Representative year of forcings | No. of ensembles | No. of modeled years in total |
|---|---|---|---|---|
| FLOR-FA | Static radiative forcing | 1860 | 1 | 3550 |
| FLOR-FA | Time-varying radiative forcing | 1861-2100 | 5 | 1200 (5 realizations of 240 years) |
| HiFLOR | Static radiative forcing | 1860 | 1 | 200 |
| HiFLOR | Static radiative forcing | 1940 | 1 | 75 |
| HiFLOR | Static radiative forcing | 1990 | 1 | 300 |
| HiFLOR | Static radiative forcing | 2015 | 1 | 70 |


With HiFLOR, there are four experiments to explore the climate sensitivity of the statistics of weather events
through static radiative forcing experiments at levels representative of particular times: preindustrial conditions (fixed at
1860 values), mid-20[th] Century (fixed at 1940 values), late-20[th] Century (fixed at 1990 values), and early 21[st] Century (fixed
at 2015 values). The value of radiative forcing agents in these experiments is prescribed from either the  CMIP5 Historical
Forcing protocol (for the 1860, 1940 and 1990 static forcing experiments) or from the CMIP5 RCP4.5 protocol (for the 2015
static forcing experiment); and the coupled atmosphere-land-ocean-sea ice state of the model is left to evolve freely. These
simulations have been integrated for different lengths of time (Table 1, last column), over which they generate their own





climate under the fixed forcing; longer integrations allow us to better estimate the statistics of climate extremes, but these
were the lengths of integrations available as of 15 August, 2016.

There are many fewer model years available with HiFLOR than FLOR-FA because the HiFLOR model was

developed more recently, and because the HiFLOR model is substantially more computationally intensive (~6× the computer
resources required for one year of integration) than FLOR-FA. The four HiFLOR static forcing experiments are initialized
from the same ocean, atmosphere, land and sea ice initial conditions, which are representative of the observed state in the
late 20th century, and the four experiments are not in radiative balance through the length of integration (the 1860
experiment has a negative top of atmosphere balance, while the 1940, 1990 and 2015 experiments have positive balances).
Therefore these static climate experiments each exhibit an initial rapid (~20 year) adjustment away from the late-20th
century observed initial conditions, and a slower climate drift reflecting the top of atmosphere imbalance over the length of
the integration. We exclude the first twenty years of each integration from our analysis, and assume (see Section 7.1) that the
impact of the slow climate drift in each model experiment on the statistics of precipitation extremes is small.

In addition to the coupled model experiments discussed above, in which the history of sea surface temperatures

(SSTs) in the models emerges from the dynamics of the models and the changes in radiative forcing, for HiFLOR a set of
variable forcing experiments were run over 1971-2015 in which the model is constrained by both historical radiative forcing
and the observed history of monthly SST (Table 2). These experiments can be used to connect the statistics of rainfall
extremes to the detailed history of SSTs that occurred over the past 45 years, part of which was a response to radiative
forcing changes and part of which emerged from internal climate variations. Furthermore by construction, these experiments
have a substantially smaller SST bias than the free running versions of HiFLOR, as the statistics of weather extremes and
their connection to larger-scale climate can be substantially affected by SST biases (e.g. Vecchi et al. 2014; Krishnamurthy
et al. 2015; Pascale et al. 2016). These experiments are described in more detail in Murakami et al. (2015) and Van der Wiel
et al. (2016). The model SST was restored to the interannually varying observed field ($SST_T$) Met Office Hadley Centre SST
product (HadISST1.1, Rayner et al. 2003)by adding an extra term to to the modeled SST tendency:
$$\frac{dSST}{dt} = O + \frac{1}{\tau}(SST_T - SST)$$   Eq. (1)
with $\tau$ the restoring time scale (three ensemble members were produced with $\tau = 5$days, three with $\tau = 10$days).

**Table 2:** Restored SST experiments performed with the HiFLOR model.

| Model | Type of forcing | Representative year of forcings | No. of ensembles | No. of modeled years in total |
|---|---|---|---|---|
| HiFLOR | Time-varying radiative forcing agents (CMIP5 Historical and RCP4.5); SSTs restored to observed monthly observations | 1971-2015 | 6 | 270 (6 realizations of 45 years) |





### 2.3 Defining an extreme event and its statistics

To classify the August 2016 south Louisiana flooding event, we must choose a definition for the event to guide our statistical analysis of observations and model experiments. We have chosen to classify extremes using multi-day averaged precipitation rather than single-day precipitation, to reflect the aspects of the event that resulted in the flooding of several rivers in the area. The following steps are taken to calculate our event statistics in the model and observations.

1. We create 3-day precipitation averages in station points/grid cells over land found in the Central U.S. Gulf Coast: 29–31 ºN, 85–95 ºW, which has relatively homogenous average precipitation extreme magnitude (Figure 1f). This provides us with, for each point in space, 365 values per year (366 in leap years) for each station point/grid cell, except the last and first years in the record when there are 364 values per year (365 in leap years), since the first January 1 and last December 31 are dropped.
2. We then, at each point in space, calculate the annual maximum for each year and define it as the local extremum for the year to create a set of extreme values for further analysis.
3. For some analyses we then take the maximum over the region. We have carefully documented in the main text when this is the case.
4. In the static forcing model experiments, we disregard the first 20 years of data to allow for some initial spin-up of the model in each new static forcing state.

In order to estimate the observed return periods using the 3-day annual events found above, we fit the resulting data to a Generalised Extreme Value (GEV) Distribution (Coles, 2001) in a similar manner as previously done for rapid attribution of the 2015 storm Desmond over the UK (Van Oldenborgh et al. 2015) and for the rapid attribution of the 2016 flooding in France and Germany (Van Oldenborgh et al. 2016). We first analyze the GEV distribution of observations and model simulations to determine if they represent the statistics of summertime extreme precipitation events sufficiently to employ them in further work. To account for possible changes due to anthropogenic climate change over time, we scale the distribution with the 4-year smoothed global mean temperature (GISTEMP for observational analysis, modeled global mean 2m air temperature for model analysis), a measure of the uniform global climate response to forcing. The GEV function is represented by:

$$F(x) = exp\left[-\left(1 + \xi \frac{x-\mu}{\sigma}\right)^{1/\xi}\right],$$         Eq. (2)

$$\mu = \mu_0 exp\left(\frac{\alpha T'}{\mu_0}\right),$$

$$\sigma = \sigma_0 exp\left(\frac{\alpha T'}{\mu_0}\right).$$

Where $\mu$ is the location parameter, $\sigma$ is the scale parameter, and $\xi$ represents the shape parameter of the curve. The ratio of $\sigma/\mu$ reduces to the constant $\sigma_0/\mu_0$. The fit is estimated using a maximum likelihood method where $\sigma, \mu_0, \sigma_0$ and $\xi$ are varied.





There is a penalty term on $\xi$: a Gaussian with a width of 0.2 is added to the likelihood function such that values larger than
~0.4 are penalized as unphysical. This is mainly used to restrain fits to the 1000-member non-parametric bootstrap that is
used to estimate uncertainty. All years are assumed to be independent for this analysis, however correlations between
proximate stations or ensemble members (when available) are taken into account with a moving block technique. The
average number of dependent stations will be noted in the analysis.

The GEV is first estimated for observational data to provide a baseline for validation. We then evaluate the

individual models by assessing the extent to which the GEV fit parameters ($\mu, \sigma$ and $\xi$) are similar to those fitted to the
longest available observational analysis (GHCN-D). As in Van Oldenborgh et al. (2016), multiplicative bias correction is
employed for the model data, which tends to  improve the similarity of the GEV fit from the model and the observations.

After a conditional GEV fit has been computed,  with global mean surface temperature as the covariate, Eq. (2) can

be inverted to find the probability of the south Louisiana event in any year. We thus estimate the probability for the south
Louisiana event in 2016, $p_1$, and its probability in some earlier year, $p_0$- taken as 1900 or the first year with available data if
that is later. The year 1900 is taken as representative for a climate that has not yet been strongly influenced much by
anthropogenic climate change. The probabilities for an event with a magnitude at least as great as that observed in south
Louisiana in each year, $i$, can be expressed as return times, $\tau_i$, by:

$$\tau_i = 1/p_i \qquad\qquad\qquad\qquad \text{Eq. (3)}$$

The ratio of probabilities or return periods from different years is known as the risk ratio where:

$$RR\ = p_1/p_0 = \tau_0/\tau_1 \qquad\qquad\qquad \text{Eq. (4)}$$

The risk ratio is a measure of how the likelihood of an event has changed in the target year (*e.g.*, 2016) versus a reference
year (*e.g.*, 1900). A *RR* value of 1 would mean that the likelihood has not changed in the baseline year versus the target year.
This ratio is therefore an indicator of changes in likelihood, but alone it cannot attribute this difference to a given
mechanism.

There are multiple methods available to evaluate the impact of radiatively-forced climate change on the change in

likelihood of events. For FLOR-FA, we repeat the analysis for the observations using data from the transient experiments.
The natural variability from an ensemble member of the model is uncorrelated with that of other ensemble members, or the
real world, so common changes in the ensemble members are therefore due to the prescribed external forcings. Multi-
decadal changes over the past century are dominated by anthropogenic forcings. For the highest-resolution global climate
model, HiFLOR, we fit a concatenated time series of maximum precipitation and the corresponding global mean
temperatures from the four static forcing experiments to Eq. (2). Furthermore, in HiFLOR we fit the trends in extremes in the
variable forcing 6-member ensemble covering 1971-2015. These simulations feature restored SSTs which reduce oceanic
temperature biases compared to a fully free running ocean component and include the same oceanic variability as the real
world (e.g. El Niño events, North Atlantic decadal variability).



We use the same procedure to investigate the effect of ENSO on extreme precipitation on the U.S. Central Gulf
Coast, replacing the smoothed global mean temperature by an index of the strength of El Niño as covariate in Eq. (2). As the
2016 flooding occurred half a year after a strong El Niño event, we take as an index a detrended version of the Niño3.4 index
with a lag of six months. The detrending is done by subtracting the average SST over 30 ºS–30 ºN.

**3 Observational analysis**

We here describe the character of the statistical distribution of observed precipitation extremes and their trends in the
GHCN-D point station data and the CPC gridded analysis by fitting to a time-dependent GEV distribution (described in
Section 2.3). Due to the many different meteorological phenomena that can lead to precipitation extremes in the Central U.S.
Gulf Coast, we assess the extent to which the GEV gives a satisfactory description of the underlying data. We frame the
results around measures of the probability per year of an event at least as intense as the 2016 south Louisiana event
(expressed as a return time), and the change of return time from the beginning of the dataset to present (risk ratio). These
return times can be assessed at a local scale (the expected wait time for an event at a particular place) or at a regional scale
(the expected return time for an event *somewhere* in the Central U.S. Gulf Coast). Because the spatial scale of the most
extreme precipitation events is substantially smaller than the whole region, the local return times are longer than the regional
return times. This observational analysis on its own is only able to detect whether a trend is present, but cannot ascribe
cause(s) to these trends. Note that from here onwards we will principally report 3-day average precipitation values rather
than 3-day precipitation sums, unless stated otherwise.

**3.1 Point station data**

We first analyze point station data, as extremes are affected by interpolation and station density, using the GHCN-D v3.22
dataset. This first analysis does not take the spatial maximum (Step 3 in Section 2.3), but analyzes all stations in the region
with at least 10 years of data. This gives 324 stations with 12536 station years with data (Figure 3a), though it is crucial to
note that they are not all statistically independent. The highest observed value at these gauges in 2016 is 216.1 mm/day at
Livingston, LA on 12–14 August (648.3 mm, three-day sum).
Fitting these data to a time-dependent GEV distribution as described in Section 2.3 gives a reasonable description of
the data (Figure 3c,e), although the fit is shaped mainly by the lower-intensity events and the highest-intensity events align
closer to the lower bound. It should be noted that for each point station in the dataset, on average another 18 are correlated
with $r > 1/e$, so the number of degrees of freedom is much less than the number of points. Overall it is surprising that all
different meteorological situations that can give rise to extreme precipitation (as laid out in Section 1) can be described with
a single GEV function.





The local return time of a 216.1 mm/day event at a station in 2016 is about 550 yr (95% Confidence Interval, C.I.,
450-1450 yr). The probability of a 3-day precipitation event at a station with 216.1 mm/day or more has increased by a factor
4.5 (C.I. 3.0-5.5) since 1900 in this analysis. This corresponds to an increase in intensity for a given return time of 22% (C.I.

16%-22%).

This fit of all data available may be influenced by the spatially and temporally varying numbers and locations of
stations. We therefore evaluate the impact of these changes in sampling on the results by limiting the analysis to stations
with at least 80 years of data and at least 0.5º of spatial separation between stations. This leaves 19 stations with 1849 station
years (Figure 3b), which results in 2.3 stations per degree of freedom on average. This analysis gives similar results: a return
time of about 500 years (C.I. 360-1400) and an increase in probability of a factor 2.8 (C.I. 1.7-3.8), corresponding to an
increase in intensity of 17% (C.I. 10%-21%), Figure 1d,f. The increase in probability is less than in the full station sample,
although compatible within the 2σ uncertainties. As the impact of inhomogeneities is smaller when considering longer time
series, we use this result from the 19 GHCN-D point stations for the trend estimate.
Our final analysis of point station data focuses on the most intense events only by considering the spatial maximum
of 3-day averaged precipitation anywhere in the Central U.S. Gulf Coast (Step 3 in Section 2.3). This answers the question
how likely an event, like that of south Louisiana 2016 or worse, was anywhere in the region, rather than at a specific place.
In the point station data, the spatial maximum is only homogeneous when the number of stations does not vary by much. We
therefore again consider only those stations with at least 80 years of data, but do not require a minimum distance this time.
The number of stations increases up to around 40 in 1950–1980 and decreases again to the present. On average 1.3 stations
are correlated at $r>1/e$ with each of these stations. We consider the period 1930-2016. The decrease in number of stations at
the end implies that a trend in extremes will be negatively biased. The number of events is lower than before (1 per year
instead of 19/324 events per year), so the uncertainties are larger.
A fit of a time-dependent GEV to the annual and spatial maximum of 3-day averaged precipitation describes the
data well (Figure 4). The return time for an event like south Louisiana 2016 anywhere in the Central U.S. Gulf Coast is
currently around 30 yr (between 11 yr and 110 yr with 95% C.I.). This is a factor 6.3 (C.I. 2.1-50) more than it was in the
climate of 1930, corresponding to an increase of intensity of about 25% (C.I. 12%-35%).
Analyses of station data analogous to the ones above but for the season July-August-September (JAS) show
somewhat smaller trends, but with larger error margins. The estimated ranges of the JAS analyses and the all year analyses
overlap.





**Figure 3**: Fit of the annual maximum 3-day average GHCN-D station precipitation on the Central U.S. Gulf Coast to a GEV that scales with smoothed global mean surface temperature. (a) Location of all GHCN-D stations with minimum 10 years of data, (c) observations (blue marks), location parameter $\mu$ (thick red line versus global mean temperature anomalies, relative to 1980-2010), $\mu + \sigma$ and $\mu + 2\sigma$ (thin red lines), the two vertical red lines show $\mu$ and its 95% C.I. for the two climates in (e). (e) Gumbel plot of the GEV fit in 2016 (red line, with 95% uncertainty estimates) and 1900 (blue line), marks show data points drawn twice: scaled up with the trend to 2016 and scaled down to 1900. The yellow square (line) denotes the intensity of the observed event at Livingston, LA. (b,d,f) as (a,c,e) but for 19 GHCN-D stations with minimum 80 years of data and minimum spatial separation of 0.5º.





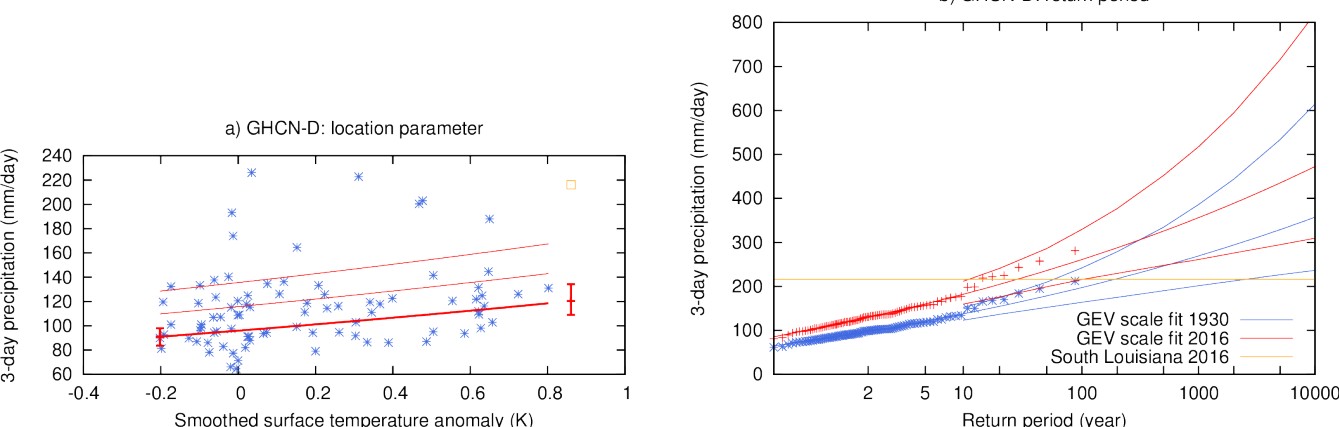

**Figure 4**: Fit of the spatial and annual maximum 3-day average GHCN-D station precipitation on the Central U.S. Gulf Coast to a GEV that scales with smoothed global mean surface temperature. (a) Observations (blue marks), location parameter $\mu$ (thick red line), $\mu + \sigma$ and $\mu + 2\sigma$ (thin red lines versus global mean temperature anomalies), the two vertical red lines show $\mu$ and its 95% confidence interval for the two climates in (b). (b) Gumbel plot of the GEV fit in 2016 (red line, with 95% uncertainty estimates) and 1930 (blue line), marks show data points drawn twice: scaled up with the trend to 2016 and scaled down to 1900. The yellow square (line) denotes the intensity of the observed event at Livingston, LA.

**3.2 Gridded analysis**

To compare with the model data, we also analysed the CPC 0.25°×0.25° gridded precipitation analysis 1948–2016. Because the spatial extent of 3-day averaged precipitation extremes is larger than the grid boxes, we first averaged these to a 0.5°×0.5° latitude-longitude grid. The highest value in 2016 is then 158.77 mm/day, which is the highest in the record. This is lower than at a single grid point due to the spatial averaging. A GEV fit of all 0.5° grid points (not shown) gives a return time of 550 yr with an uncertainty from 300 to 2000 yr, compatible with the station analysis but with larger uncertainties. The probability has increased by a factor 3.5 (C.I. 2.0-11) since 1948, corresponding to an increase in intensity of 15% (C.I. 9%-24%).

Taking the spatial maximum of the original 0.25°×0.25° grid we find that the highest observed value in 2016 is 178.2 mm/day on 12–14 August (534.7 mm in three days). The record is too short to draw robust conclusions from a fit of a GEV depending on global mean temperature except that the precipitation maxima also increase in this dataset (Figure 5). In this dataset, the return time for an event like 2016 anywhere on the Central U.S. Gulf Coast is currently between 9 and 200 yr (best estimate 25 yr). This is about a factor 5 (C.I. 1.1-60) larger than it was around 1948, which equates to an increase in intensity for an event like 2016 of roughly 15% (C.I. 0.4%-30%).

As for station data, analyses of CPC similar to the ones above but for the season JAS show somewhat smaller trends, but with larger error margins. The estimated ranges of the JAS analyses and the all year analyses overlap.





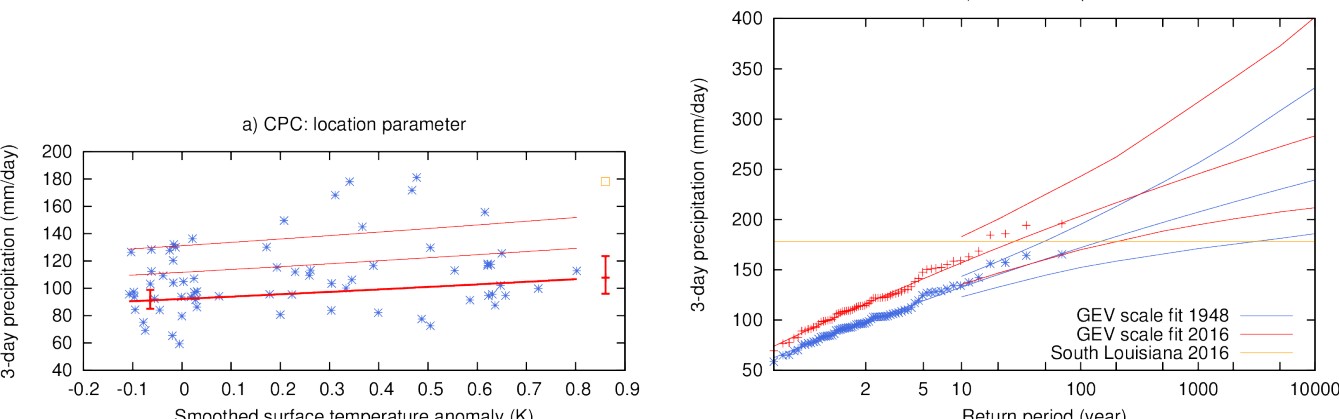

**Figure 5:** As Figure 4 but for the spatial and annual maximum 3-day average 1948–2016 0.25º×0.25º gridded CPC analysis.

### 3.3 Influence of natural variability

We investigate the influence of natural variability on the probability of an event like south Louisiana 2016 by using indices of detrended SST as covariates in the time-dependent GEV fits. We first examine the influence of El Niño-Southern Oscillation (ENSO) by using as a covariate 6-month lagged Niño 3.4-index (5 ºS–5 ºN, 170–120 ºW) minus SST averaged of 30 ºS–30 ºN to remove to first order the effects of global warming. This is inspired by the heavy rain events after the 1997/98 El Niño event. A comparison of recent Niño 3.4 conditions with those from a year following the strongest La Niña year (1917) in a fit of all 324 stations with more than 10 years of data suggests that anomalously warm tropical Pacific SSTs significantly ($p < 0.1$) increase the probability of an event like south Louisiana 2016, but not by much. In the year after El Niño, the probability is a factor 1.3 (C.I 1.0-1.9) higher than in a year following a very strong La Niña. However, the maximum of stations with at least 80 years, which represents the largest events, does not show a signal, albeit with a large uncertainty of a factor 0.5 decrease to a factor 1.7 increase.

  Simultaneous correlations with global SSTs indicate a region in the North Atlantic that has a significant relationship with Central U.S Gulf Coast extreme precipitation at p<0.1 (Figure 6). Although the field significance is very low, the region is a well-known source of decadal variability and predictability (e.g., Hazeleger et al. 2013), so we still consider it a possible source of decadal variability of extreme precipitation. We use an area-average of SSTs between 45–60 ºN and 50–20 ºW as a covariate in the GEV fit. The region was anomalously cold in 2016, so we compare the changed probability with a warm year (2006). In this statistical analysis, North Atlantic SSTs are significantly correlated ($p < 0.01$) to Central U.S Gulf Coast precipitation (by design, as we chose the region that has a significant correlation), with recent below average SSTs decreasing the probability of an event like 2016 (risk factor 0.37, C.I. 0.11-0.81). To ascertain whether this is a physical connection and not just a coincidence by picking the region of largest correlations, we need to analyse model results.



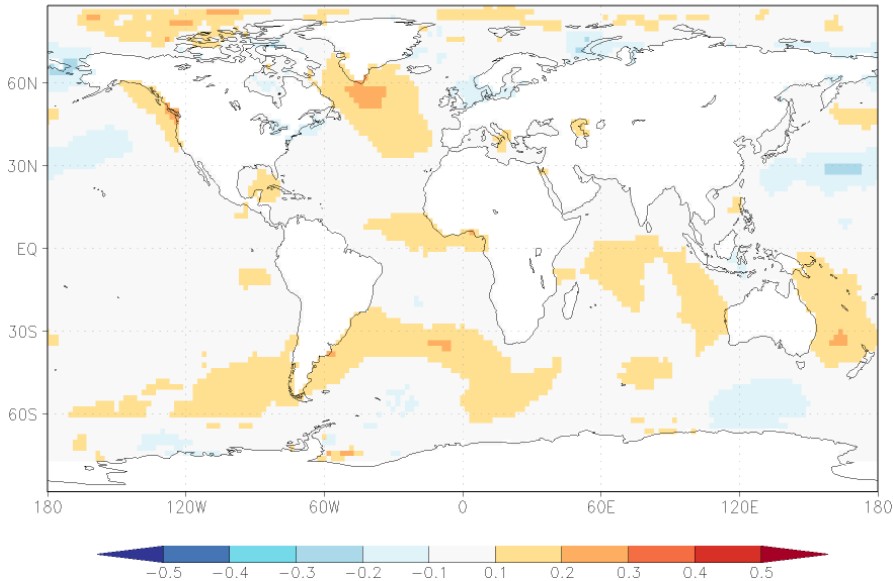

**Figure 6:** Correlation coefficient between Central U.S. Gulf Coast spatial and annual maximum of 3-day extreme precipitation intensity and annual mean SST (ERSST v4) with a linear regression on the global mean temperature removed at each grid point.

## 4 Model evaluation

We here describe an evaluation of simulated precipitation extremes in the two global coupled models (model descriptions in Section 2.2). Precipitation is a notoriously difficult field to simulate, as many coupled climate models exhibit large biases (Dai 2006, Flato et al. 2013). Though FLOR-FA and HiFLOR underestimate the intensity of Central U.S. Gulf Coast precipitation extremes slightly, this bias is significantly reduced in these high-resolution models compared to standard-resolution models (Van der Wiel et al. 2016).

### 4.1 Annual cycle and intensity

First we analyse the annual cycle of extreme precipitation intensity. We consider the median and 97.5 percentile of the monthly maximum of the spatial maximum of 3-day averaged precipitation (Figure 7). The 97.5 percentile events are of smaller magnitude than the south Louisiana observed event (100-150 mm/day versus 200 mm/day), but we consider smaller magnitude events to increase the number of events in the calculation and hence decrease uncertainties.

The observed precipitation extremes in spring and summer are generally more intense than in autumn and winter (Figure 7a). There is no agreement between the two observational products on which season sees the most intense precipitation extremes (97.5 percentile, Figure 7b), though extremes in March-October are more intense than in winter. This





period of stronger extremes is longer than the hurricane season, which provides a fraction of these extremes. In this region,
the models underestimate the intensity of extreme precipitation, which was also noted in Van der Wiel et al. (2016). FLOR-
FA has a peak season for extreme precipitation intensity in JAS which is not found in the observational data. The HiFLOR
SST-restored experiment, in which global SST biases are decreased compared to the free running experiments, shows a
similar peak in JAS. The HiFLOR 1990 static forcing experiment however, doesn't show this peak. Instead it has a similar
annual cycle structure to the observational data, though with a smaller amplitude.

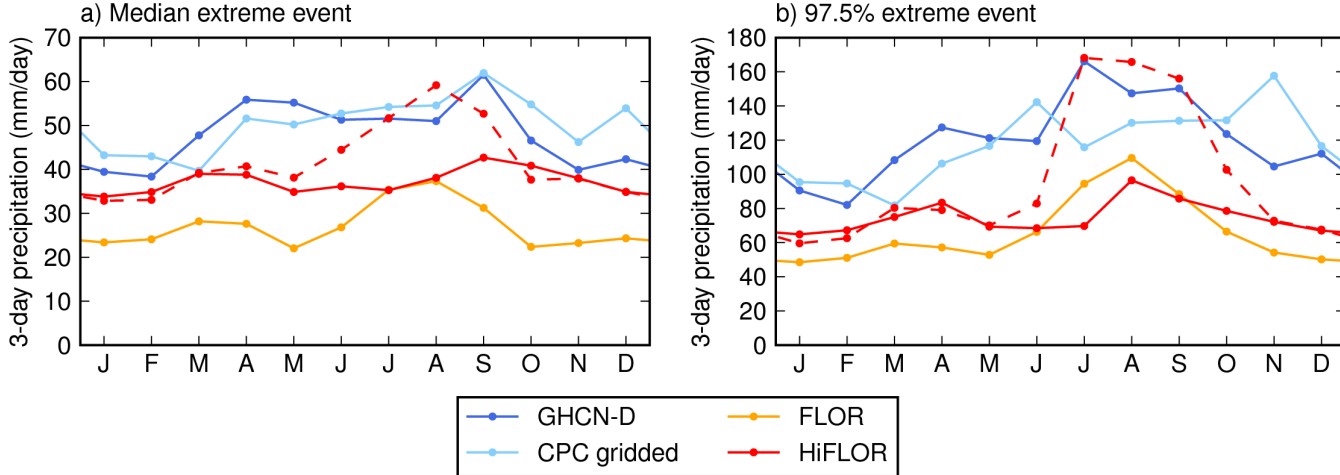

**Figure 7.** Annual cycle of monthly and spatial maximum 3-day averaged precipitation for point station data (GHCN-D, dark
blue line), gridded observational data (CPC, light blue line) and model simulations (FLOR-FA, orange line, and HiFLOR,
red lines). For HiFLOR the 1990 static forcing experiment (solid red line) and the variable forcing SST-restored experiment
(dashed red line) are included. Shown are (a) the median value of the monthly extremes and (b) the 97.5 percentile.

**4.2 Meteorological conditions**
Next, we investigate the meteorological conditions generating extreme precipitation events in both models and compare
these to the observed ones. For this analysis we consider the longest static forcing experiments for each model: 1860 for
FLOR-FA and 1990 for HiFLOR and the CPC gridded precipitation analysis. The selection of these events is limited to the
region of interest (Central U.S. Gulf Coast) and the months JAS to facilitate comparison against the south Louisiana event.
Precipitation totals and circulation patterns for the nine largest extreme precipitation events in the CPC analysis
(JAS season only) are shown in Figure 8. Note that the 2016 south Louisiana event ranks as number 2- heavy precipitation
related to Hurricane Danny in 1997 was stronger, though it was confined to a smaller area. Seven of these nine events were
associated with a tropical cyclone/hurricane making landfall (78%, orange tracks are the International Best Track Archive for
Climate Stewardship, IBTrACS, track estimate, Knapp et al. 2010), the exceptions are July 1975 and, as noted before,
August 2016. Note that the GEV analysis in Section 3.2 was based on annual maxima, for which the ranked extreme events



are different than the ones shown in Figure 8 (these are nine of the top 14 events when all data is taken into account, ranks 1 and 2 are the same).

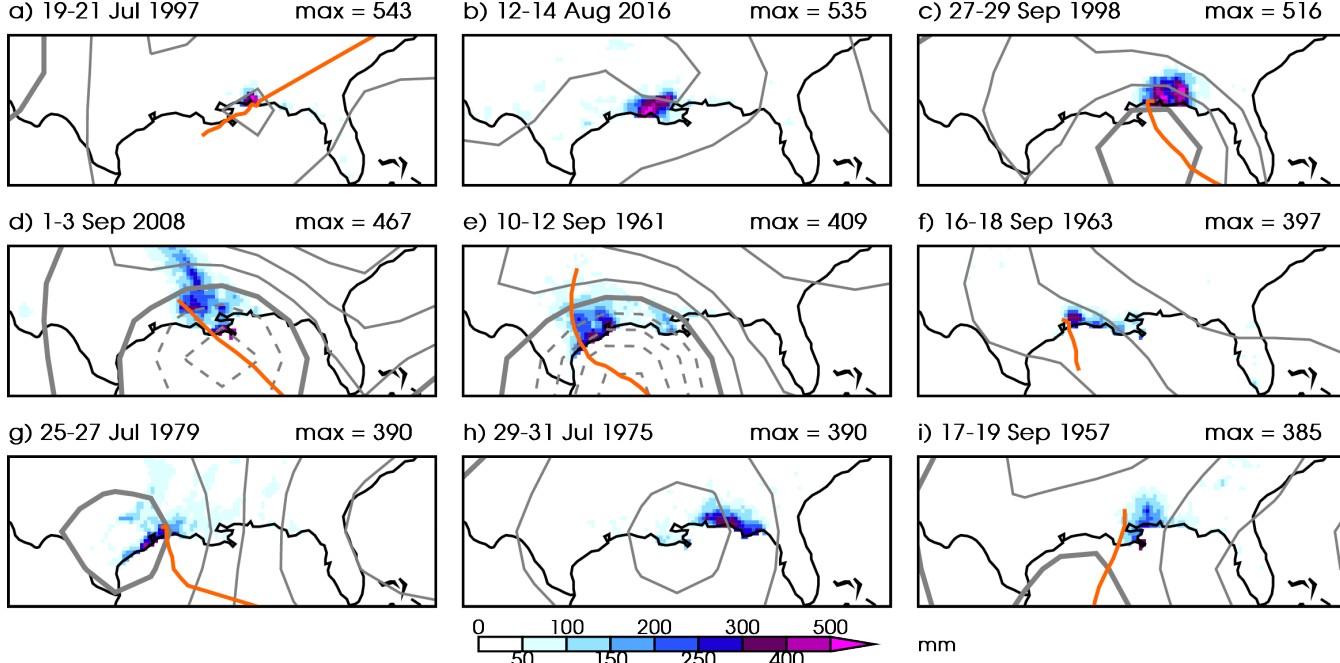

**Figure 8:** Top 9 extreme precipitation events in the Central U.S. Gulf Coast (29–31 ºN, 85–95 ºW) for the CPC gridded precipitation analysis. 3-day precipitation sum (mm, shaded colors, as in Figure 1d), 850-hPa height for the middle day (grey contours, interval 25 m, 1500 m contour thickened, lower contours dashed) from NCEP/NCAR Reanalysis 1 (Kalnay et al. 1996) and tropical cyclone track if system is classified as one (orange line, IBTrACS). These extreme events are calculated for the three month period: JAS.

A similar figure for FLOR-FA is included as Figure 9. We now show the 18 most extreme events (approximate return period 3530/18≈200 years) in FLOR-FA. The return period in the model for these events is much larger than the return period for the observed events in the CPC analysis (approximate return period 69/9≈8 years). Despite the negative bias of precipitation extreme intensity (Section 4.1), the precipitation sums for these events are therefore larger than those in the observed data. All events are associated with a low pressure system, of which 8 (44%, orange tracks in Figure 9) are a tropical cyclone based on the TC tracking methodology of Harris et al. (2016) as implemented in Murakami et al. (2015). Note that the low pressure systems of the top 4 events do not classify as a tropical cyclone, showing the precipitation potential of non-tropical cyclone low pressure systems in the model.





**Figure 9:** As Figure 8 but for the top 18 maximum extreme precipitation events in the 1860 FLOR-FA static forcing experiment. Note that years are model years and do not resemble dates on the real world calendar and that the model provides precipitation information over ocean grid boxes too.



Because the HiFLOR 1990 static forcing experiment is of smaller length, it is not possible to sample the 200-year return period event as was done for FLOR-FA adequately. In Figure 10 we show the 6 most extreme events (approximate return period 280/6≈50 years, the top 2 events are samples of events with return periods of about 150 years). In HiFLOR the most extreme precipitation events are the result of a tropical cyclone, though storm intensity (storms in Figure 10a,b are tropical storms, storms in Figure 10c,d are hurricanes at the time of landfall) is not related to resulting precipitation magnitude. Note that the strongest event in HiFLOR exceeds 900 mm over a 3-day period, which is much stronger than the observed values in south Louisiana.

In conclusion, though the precipitation extremes are of smaller magnitude in both models and the annual cycle in observations is not recovered well (Section 4.1), the meteorological system leading to these precipitation extremes in JAS are realistic and resemble observed systems (Section 4.2).

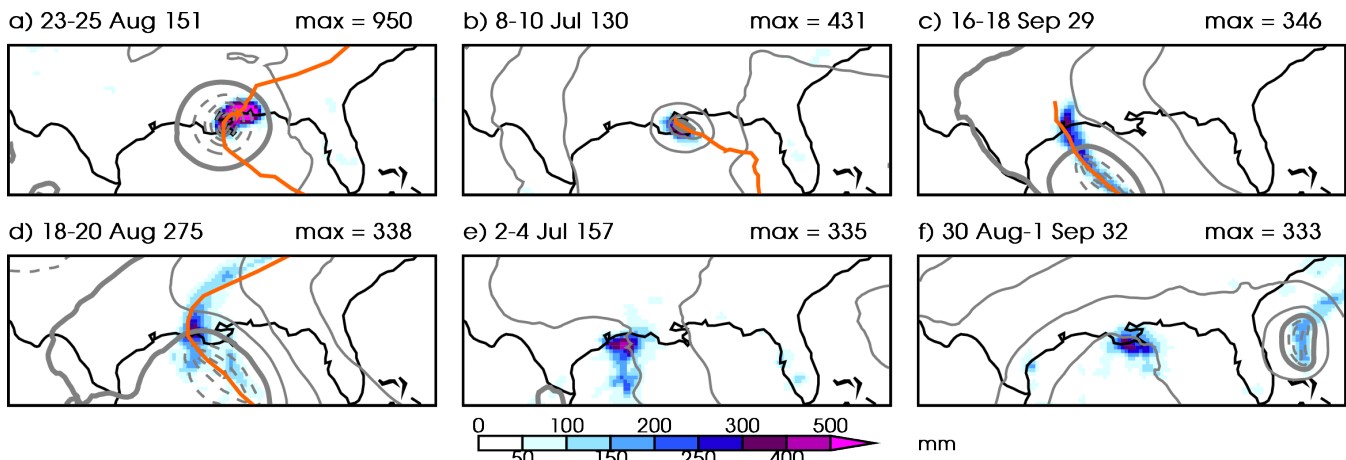

**Figure 10**: As Figure 8, but now for the top 6 maximum extreme precipitation events in the 1990 HiFLOR static forcing experiment. Note that years are model years and do not resemble dates on the real world calendar and that the model provides precipitation information over ocean grid boxes too.

## 5 Model analysis

In order to attribute the observed trend to external forcing we use global climate models that isolate the different forcings. The model and experimental description can be found in Section 2.2.

## 5.1 FLOR-FA

A fit of all land grid boxes (0.5°×0.5°, 23095 data points) to a time-dependent GEV distribution is shown in Figure 11. The uncertainties take into account the dependencies by moving spatial blocks of 7.7 grid points on average. In contrast to the observations (Figure 3) the distribution cannot be described with a single GEV function: the extremes with return times



larger than about 100 years (80 mm/day) diverge from the fit that is determined mainly by the less extreme precipitation
events. This so-called 'double population' problem results from different meteorological mechanisms for extreme events. We
therefore cannot use this fit for attribution.

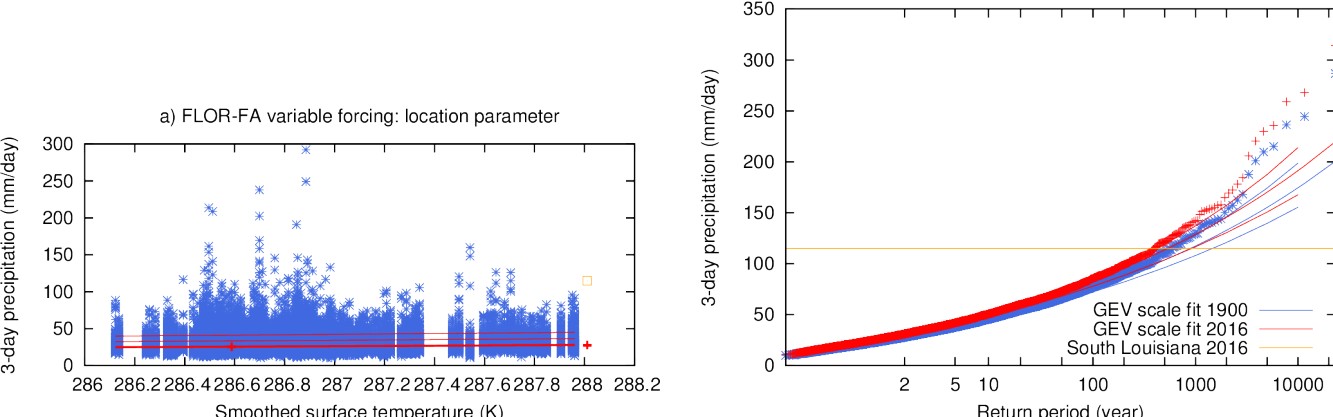


**Figure 11**: As Figure 4 but for the annual maximum 3-day average precipitation in the FLOR-FA variable forcing
experiment (based on complete experiment, 1861-2100).

Taking the spatial maximum of all grid boxes selects only the high end of the distribution. Figure 12a,c shows the
GEV fit to these extremes using data for simulated years 1861-2015. The fit is still not completely satisfactory as the highest
five events (all in the early years of the experiments) fall on the upper boundary of the 95%C.I. around the fit to the rest of
the distribution. Due to this, the shape parameter $\xi$ and scale parameter $\sigma$ of the GEV distribution are higher than they are in
the observations. Because of model bias, we adjust the model amplitude of extremes to obtain the same return time as that in
observational data, of around 30 years (115 mm/day). This gives a trend in this model that is significantly greater than zero
at $p<0.05$ (one-sided). However, the factor 1.3 (C.I. 1.0-1.9) increase in probability, corresponding to an increase in intensity
of 5% (C.I. -1%-14%), is much less than the observed one .
Assuming that the relationship with global mean surface temperature does not change in the model world until
2100, in spite of a different mix of anthropogenic forcings (greenhouse gases and aerosols), we can improve the signal-to-
noise ratio of the fit by using all data in the variable forcing experiment (Figure 12b,d). For the spatial and annual maximum
of 3-day averaged precipitation this gives an increase in probability of a factor 1.8 (C.I. 1.4-2.0) corresponding to an increase
in intensity of 11% (C.I. 7%-12%) up to now.
Analogous analyses but for the season JAS show similar results, although with larger error margins. We looked for
an effect of ENSO in the long static forcing experiment in the same way as in the observations. This does not show any
influence of El Niño averaged over the 12 months July–June preceding the year of extreme precipitation events.





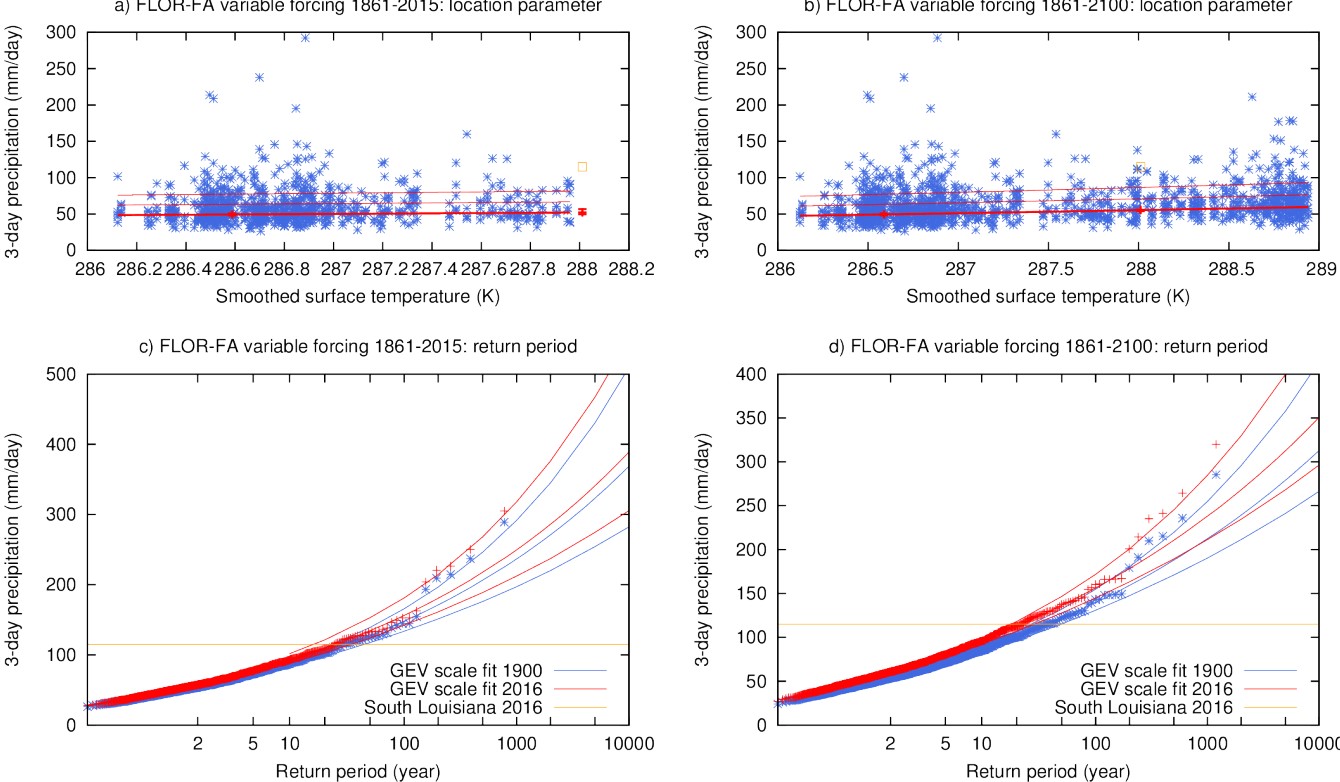

**Figure 12**: As Figure 4 but for the annual and spatial maximum 3-day average precipitation in the FLOR-FA variable forcing experiment. (a,c) taking into account years 1861-2015, (b,d) taking into account 1861-2100.

## 5.2 HiFLOR

The HiFLOR model at a higher 25 km resolution has a more realistic seasonal cycle, but underestimates extreme precipitation by 25% for a 1 in 1 year event and by 35% for 1 in 1000 year extremes. We correct for this bias by defining our event to have the same return time as the gridded observations in 2016, that is, 103 mm/day. We concatenated the four static forcing experiments that we have available, leaving out the first 20 years of each, to create a 655-year record. To decrease dependencies we averaged 2×2 grid boxes into a 0.5º grid, this results in each grid box being correlated with 10.3 others with $r>1/e$ on average.

As was found for FLOR-FA, the GEV fit to all grid points results in a double population, therefore we disregard that analysis and instead focus on the spatial maximum precipitation extreme. Similar for FLOR-FA, taking the spatial maximum of this 50 km dataset selects mainly events in the more extreme population and does give a good fit to the GEV distribution (Figure 13). The outlier event is a tropical cyclone in the 1990 static forcing event, that was discussed in Section 4.2 (Figure 10a). The external forcing, which is the only change between the static forcing experiments, causes an increase in





probability of a 103 mm or stronger event of a factor 2.0 (C.I. 1.4-2.5), in agreement with the FLOR-FA experiment up to
2100 (Figure 12b,d). This corresponds to an increase in intensity of 10% (C.I. 5%-12%).

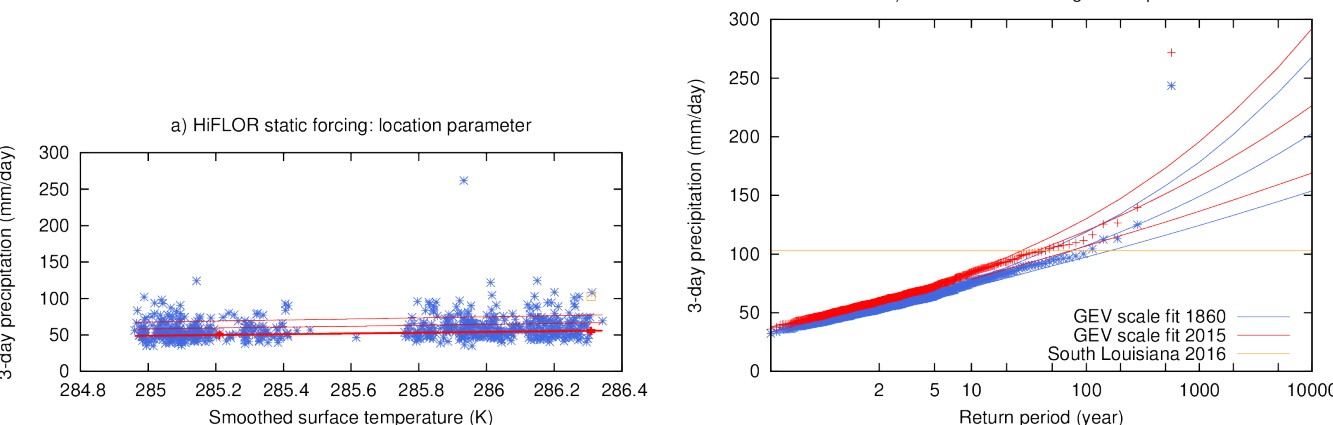

**Figure 13**: As Figure 4 but for the annual and spatial maximum 3-day average precipitation in the HiFLOR static forcing
experiments.

An analysis of these data using the annual averaged detrended Niño3.4 index lagged by 6 months as covariate
shows a relatively strong influence of El Niño in this model, with an increase in probability from the year following
strongest La Niña to the strongest El Niño of a factor about 4.2 (C.I. 1.7–6.7).
We followed the same procedure on the six ensemble members of the variable forcing HiFLOR experiment (1971–
2015). These simulations do not have a negative bias in extreme precipitation. The restored SSTs eliminate a 2 K cold bias in
the subtropical Atlantic that is present in the static forcing experiments, which may have caused the bias in precipitation
extremes on the U.S. Central Gulf Coast in those simulations. Again there is one outlier event with 452.8 mm/day over three
days, 1351 mm total.
The spatial and annual maximum of 3-day averaged extreme precipitation increases by a factor 1.8 (C.I. 1.2–3.3) in
these experiments over the period 1971–2015, corresponding to a change in intensity of 14% (C.I. 4%–27%), Figure 14.
Although the restoring of SSTs increases the fidelity of the simulation, it also includes the non-forced natural variability of
the real world, so these numbers do not isolate the forced change but show the full change including the effects of natural
variability. Assuming these are small compared to the trend we can extrapolate to the full change since 1900; the period
1971-2015 only includes about 2/3 of global warming since preindustrial times. This translates to a factor 2.4 (C.I. 1.3–6)
increase in probability and 22% (C.I. 6%–41%) in intensity, which is very similar to the trend found in the observational
data.
Analyses of the season JAS show similar to somewhat smaller trends, but with larger error margins, overlapping the
all-year error margins.





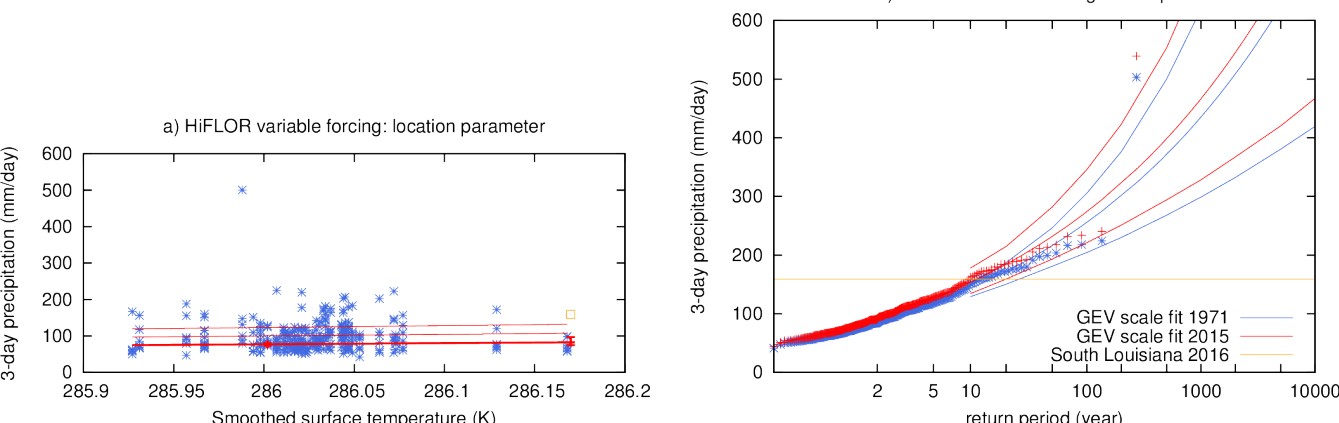

**Figure 14**: As Figure 4 but for the annual and spatial maximum 3-day average precipitation in the HiFLOR variable forcing restored SST experiments.

## 6 Summary

In this section we summarize the principal observational and model-based results as described in Sections 3 and 5. We have analyzed two observational data products (GHCN-D point station data and CPC 0.25°×0.25° gridded analysis), to estimate the probability, and changes in probability and intensity of a 3-day precipitation event as large as that observed in south Louisiana 2015. The analysis was confined to the Central U.S. Gulf Coast (29–31 ºN, 85–95 ºW) and relies on time-dependent GEV fits to the data. First we investigated probabilities and changes at a single station, i.e. the probability of such an event *at a fixed place* in the region. Second we investigated regional probabilities and changes, i.e. the probability of such an event *anywhere* in the region. The spatial scale of the most extreme precipitation events is significantly smaller than the region considered, therefore the second probability is lower than the first. To attribute the observed changes to forced anthropogenic climate change, we repeat the analysis using high-resolution global climate model data from GFDL FLOR-FA and GFDL HiFLOR. GEV fits for the local analysis were unsatisfactory, therefore we only report the regional change in probabilities.

The expected return period of a comparable 3-day precipitation event at a single station as high as the maximum observed is 450 to 1450 year, best estimate 550 year. Return periods like these are often written as a "1 in 1000 year event". The return time for observing an event anywhere in the region is lower: between 11 and 110 year. All observational analyses found clear positive trends, with an increase in probability for the regional event of about a factor 6.3 (97.5% certain more than 2.1), and an increase in intensity of 12% to 35% (Table 3). Estimates based on CPC gridded data are comparable but have larger ranges due to the shorter period of data availability.



**Table 3**: Summary of observed (first two rows) and modeled (third row and down) changes in regional rainfall extremes in Central U.S. Gulf Coast.

| Data source (years used for calibration) | Baseline regional return period for 2016 event (95% confidence range, observations only) | Years change calculated over | Change of return period in present day over given years (95% confidence range) | Change in intensity of regional 30-year return event in 2016 since beginning of record (95% confidence range) |
|---|---|---|---|---|
| GHCN-D rain gauges, minimum 80 year data (1930-2016) | 30 year (11 - 110) | 1930-2016 | 6.3× (2.1 ... 50) | +25% (12% ... 35%) |
| CPC 0.25°×0.25° gridded data (1948-2016) | 25 year (9 - 200) | 1948-2016 | 5.4× (1.1 ... 60) | +15% (0.4% ... 30%) |
| FLOR-FA variable forcing experiment (1861-2015) | | 1900-2016 | 1.3× (1.0 ... 1.9) | +5% (-0.5 ... 14%) |
| FLOR-FA variable forcing experiment (1861-2100) | | 1900-2016 | 1.8× (1.4 ... 2.0) | +11% (7% ... 12%) |
| HiFLOR static forcing experiment (1860, 1940, 1990, 2015) | | 1860-2015 | 2.0× (1.4 ... 2.5) | +10% (5% ... 12%) |
| HiFLOR variable forcing experiment (1971-2015), extrapolated to 1900-2015 | | 1900-2015 | 2.4× (1.3 ... 8) | +22% (6% ... 41%) |

The sensitivity of precipitation extremes from both models is consistent with that estimated from the gridded observations. The lower-resolution FLOR-FA model shows lower trends than the HiFLOR model. For the HiFLOR model the sensitivity estimated from the SST-restored experiment for 1971–2015 is larger than that from the coupled simulations. Taking into account all modeling results, the probability of an event like south Louisiana 2015 has increased at least by a factor 1.4 due to radiative forcing; the two HiFLOR experiments and the analysis of the full dataset from FLOR-FA suggest central values close to a doubling of probability. Such an increase may be translated to what was once a 1/100 year event somewhere in the Central U.S. Gulf Coast, should now be expected to occur on average, at least once every 70 years, likely even more common. This trend is expected to continue over the 21st century as past and projected future greenhouse forcing continues to warm the planet.

The evidence for an influence of the strong 2015/2016 El Niño increasing the probability of the 2016 event is equivocal. The full station dataset shows a statistically significant but small increase in probability, but we do not find the same for the spatial maximum, which represents the strongest events. The FLOR-FA model similarly does not have an ENSO effect, whereas the HiFLOR model again shows a higher probability after a large El Niño. We have found some



evidence for decadal Atlantic variability affecting precipitation in the observations, which would have decreased the
likelihood in 2016 if confirmed.

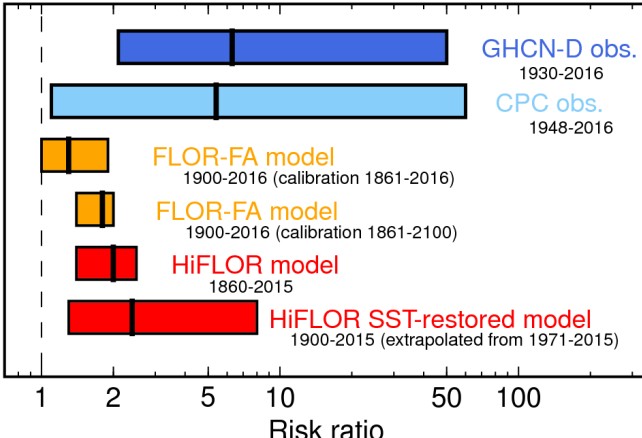
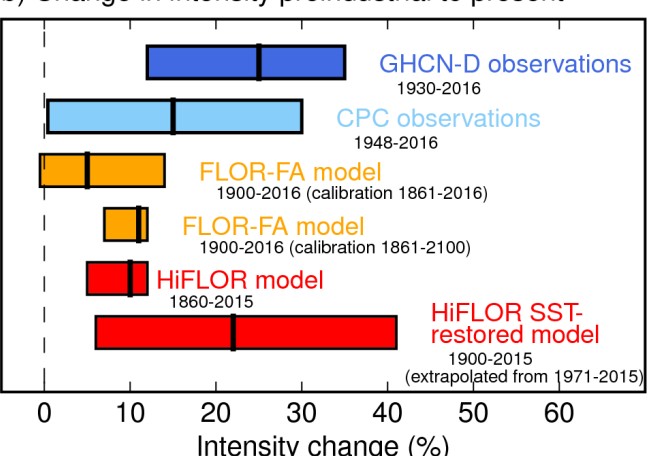


**Figure 15**: Summary of observed (GHCN-D, CPC, blue colors) and modeled (FLOR-FA, HiFLOR, yellow, red color)
changes in regional precipitation extremes in Central U.S. Gulf Coast. Ranges written in black are the time periods for which
the change is shown over. Calibration for the calculations is done over separate time periods for noted models. See Table 3
for specific numeric values.
**7 Discussion**
We have presented a rapid attribution to climate change and climate variability of the south Louisiana intense precipitation
event. Here we lay out the crucial assumptions made to conduct our assessment, further lines of inquiry to investigate the
validity of the crucial assumptions and the sensitivity of our results to changes in these assumptions, suggestions for further
study on related topics not investigated here, and questions that arise from this work. Finally, we note some societal impacts
and management implications of the findings.
**7.1 Crucial assumptions**
In performing these analyses, we have made the following crucial assumptions about the observations, models, the statistical
distribution of precipitation extremes, and the relationship between temperature and precipitation extremes. We have tested
the sensitivity of our results to some of these assumptions in the results sections (Sections 3-5) and discuss them below.

1)   We assume that the local, annual maxima of 3-day averaged precipitation over the region of analysis (29–31 ºN,

85–95 ºW) can be grouped together, and that their statistical distribution follows a GEV distribution. Underlying

this assumption is that the region has homogeneous extreme precipitation characteristics (Figure 1f). Furthermore,





we assume that all the annual maxima of 3-day averaged precipitation are drawn from the same statistical distribution, in spite of the many different mechanisms that lead to extreme precipitation in this region, and that this distribution can be represented well by a GEV distribution. We further assume that the spatial maximum over the region can also be described by a GEV.

2) We assume that analyzing all seasons together provides a fuller distribution of the population of extreme precipitation events than isolating the analysis to seasons proximate to August (the month in which the south Louisiana event occurred). In part, the choice to analyse annual extreme events was motivated by the fact that a variety of meteorological phenomena can lead to extreme precipitation in this region, flooding can occur in any season, and precipitation extremes may change in various seasons (Lehmann et al. 2015, Van der Wiel et al. 2016). All extreme value analyses were repeated focusing only on the JAS season and the qualitative nature of the results was the same as those presented.

3) We assume that the inhomogeneities in point station data due to station changes, incomplete records and geographic coverage are smaller than the trends and have no coherent sign. We have checked this by performing the analysis on all stations and for a subset of stations with long (at least 80 year) records and sufficient (0.5º) spatial separation.

4) We assume that the methods that create the gridded observationally-based precipitation data result in an accurate representation of 3-day average precipitation at the grid scale. The decorrelation scale of 3-day precipitation is about twice the grid scale, so the largest uncertainty is the inhomogeneous distribution of the gauge stations in space and time. A comparison of the results with point station data shows that the differences are not large.

5) We assume that, for the assessment of trends in GEV statistics, global mean surface temperature represents a relevant covariate to capture the *a priori* expected connection between precipitation extremes and temperature (e.g., O'Gorman 2015). A physical motivation for this expected connection is the dependence of the saturation specific humidity of air on temperature through Clausius-Clapeyron (see Section 1). The underlying assumption is that multi-decadal temperature changes exhibit "pattern scaling", such that global mean temperature change is a sufficient parameter to describe the long-term changes of temperature; furthermore, global-mean temperature helps increase the signal-to-noise ratio of fits to temperature changes. If there is substantial spatial heterogeneity to temperature changes on multi-decadal timescales, the assumption that global mean temperature is the relevant metric becomes suboptimal. Furthermore, if dynamical changes (e.g., changes in the statistics of storms, changes in the dominant moisture sources for extremes, etc.) dominate the observed multi-decadal precipitation extreme changes, this assumption will also be suboptimal.

6) We assume that the probability density function of precipitation extremes scales with a covariate, for example (smoothed) global mean temperature and does not exhibit other changes in shape. This assumption is supported by large-sample statistics from modelling experiments such as Weather@Home (Massey et al. 2015) in other regions, but it is not *a priori* obvious that these results should also hold for the Central U.S. Gulf Coast with its wide variety





of weather phenomena causing extreme precipitation. Furthermore, the Massey et al. (2015) results were from models of resolution too low to resolve many of the meteorological phenomena that lead to extreme precipitation (e.g. tropical cyclones) in this region.

7) We assume that, beyond an initial rapid (~20 year) adjustment to different static radiative forcings, the statistics of precipitation extremes in the static forcing model experiments depend on global mean temperature in the same way as the changes arising from slow drift due to top of the atmosphere radiative disequilibria and slow ocean adjustment. The latter changes are smaller than the forced trend, so the impact of slow model drift on the results is small.

8) We assume that the CMIP5 historical forcings (1860-2005) and RCP4.5 forcings (2005-2100), as implemented in the models, are sufficiently accurate representations of the actual changes in radiative forcing that occurred in the real climate system to allow meaningful comparison of modeled changes in precipitation extremes to those observed.

9) We assume that the FLOR-FA and HiFLOR modeled responses to changes in radiative forcing are meaningful estimates of the sensitivity of precipitation extremes in the real climate system, since these models capture multiple physical factors affecting precipitation extremes in a physically-based and internally-consistent framework. This assumption is motivated in part because of the ability of these models to simulate large-scale precipitation and temperature over land (e.g., Van der Wiel et al. 2016; Delworth et al. 2015; Jia et al. 2015, 2016), precipitation extremes over the U.S. (Van der Wiel et al. 2016), modes of climate variability (e.g., Vecchi et al. 2014; Murakami et al. 2015); the meteorological phenomena that lead to precipitation extremes and their relationship to modes of climate variability (e.g., Vecchi et al. 2014; Krishnamurthy et al. 2015; Murakami et al. 2015, 2016; Zhang et al. 2015, 2016; Pascale et al. 2016); and that these models show skill at seasonal predictions of large-scale climate, regional hydrometeorology and the statistics of weather extremes across a broad range of climatic regimes (e.g., Vecchi et al. 2014; Jia et al. 2015, 2016; Yang et al. 2015; Msadek et al. 2015; Murakami et al. 2015, 2016). However, it is important to note that climate models can show a range of global and regional climate sensitivities to changing radiative forcing (e.g., Kirtman et al. 2013, Collins et al. 2013)

These assumptions were crucial to enable a rapid assessment of the climate context of the extreme precipitation of the August 2016 south Louisiana event. Subsequent analyses should further assess the validity of these assumptions, and the quantitative impact of failures in their validity. Below we outline our present evaluation of the implications of these choices and potential areas of further research.

Sensitivity experiments should be produced by varying the parameters of our study. We did not conduct analysis of how the size of our defined box for the Central U.S. Gulf Coast affects our results (crucial assumption 1). If the region is altered to remove points that have greater risks relative to those included, the findings may change. Changes in extreme





precipitation  risks in the Central U.S. Gulf Coast should not be applied elsewhere without further investigation. Temporally,
we were able to validate the seasonal distribution of precipitation extremes in models and observations (Section 4.1), and
redid the analysis for JAS only, which gave larger uncertainties and somewhat smaller trends (crucial assumption 2). Future
work could further quantify seasonal differences in extremes and their response to climate forcing. Similarly, to sample the
spread in sensitivity to future RCP forcings (crucial assumption 8, used for any modeled years beyond 2005), our results may
be revised with different climate forcings. For the near term however, this is likely not an issue in HiFLOR (used to produce
climates for 2005-2015 in the static forcing and nudged SST runs) as climate variability tends to be greater than the climate
response to different scenarios during this time period  (Forster et al. 2013; Hawkins and Sutton 2009; Kirtman and Power
2013), but may affect future climate results in the FLOR-FA variable forcing experiment at the end of the century (2100,
Hawkins and Sutton 2009). Finally, the appropriateness of GEV fits in general should be tested (crucial assumptions 1,6).
Sensitivity experiments of our results to model bias and integration length (or length of the observed record) should
be produced (crucial assumptions 3 and 7). Short records limit the reliability of the statistics of precipitation extremes. This
is important for our model validation of the annual cycle of extremes (Section 4.1) and for the comparison of modeled and
observed GEV fits (Section 5). The statistics of precipitation extremes in HiFLOR are closer to those observed than the
statistics in FLOR-FA. However, we note that the model experiments with FLOR-FA are significantly longer and therefore
provide better statistics of its (biased) climate than the experiments with HiFLOR or the observed record. It cannot thus be
fully-excluded that the double distribution of extremes in FLOR-FA or the large peak in JAS in extreme precipitation
intensity is purely a result of model bias.
A portion of the beginning of the static forcing experiments have been disregarded to allow the model to spin-up in
response to radiative forcing. GEV fits were originally calculated by disregarding the first 10 years of data to allow for spin-
up, but was extended to 20 years to provide the simulated climate more time to approach equilibrium (crucial assumption 7).
The results are only altered slightly by this sensitivity test. Given the length of the available ensemble suite of static forcing
experiments, disregarding more years in the beginning of the simulation would reduce our ability to sample extremes. With
longer integrations of static forcing experiments and additional ensemble members, we would have more information to
assess how model spin-up may affect our results. Similarly, longer integrations would allow for an assessment of the impact
of model drift due to ocean adjustment (crucial assumption 7).
The attribution to climate change presented here depends on our assumption that changes in precipitation extremes
scale with global mean temperature and do not arise from changes in the shape of their underlying distribution (crucial
assumptions 5 and 6). The thermodynamic basis of this assumption is based on a large body of research (O'Gorman 2015),
however as noted before there is a large variety of synoptic systems that may cause precipitation extremes in the Gulf Coast
region. It is not obvious that possible impacts of changes in synoptic weather patterns scale with global mean temperatures.
For example, the frequency, track location and/or intensity of tropical cyclones (responsible for 7 out of the 9 most extreme
events in JAS were related to tropical cyclones, Figure 8) can each change in complex ways that need not scale with each



other or global mean temperature (e.g., Vecchi and Soden 2007; Murakami and Wang 2010; Emanuel and Sobel 2013; Emanuel et al. 2013; Knutson et al. 2013; Vecchi et al. 2013; Walsh et al. 2015), and could cause changes to the statistics of extreme rainfall in the Central U.S. Gulf Coast. Further research must investigate what the impact of dynamic changes (e.g. frequency of occurrence of various synoptic systems, dominant moisture sources, precipitation efficiency) is on the presented trend of precipitation extremes.

To investigate the sensitivity of the results to the chosen observational data sets (both based on rain gauge measurements, crucial assumption 3 and 4), we suggest repeating the current analysis with an independent observational estimate of current and historical precipitation along the Gulf coast (e.g. estimates based on satellite data). Furthermore, though we use two global climate models (FLOR-FA and HiFLOR, crucial assumptions 7 and 9) and various experimental setups (static radiative forcing, time-varying radiative forcing and restoring observed SST variability), the models are part of the same NOAA/GFDL family. Consequently, they exhibit similar patterns of (surface temperature) bias and rely on the same parameterization schemes for precipitation. Further inquiry for understanding model-specific biases that may impact the results may still be warranted. For example, there is a North Atlantic cold bias in the models, thought to be connected in part to inadequate eddy parameterizations and a resulting cloud feedback (Delworth et al. 2006; Delworth et al. 2012; Vecchi et al. 2014; Murakami et al. 2015). This may be the source of higher magnitudes of modeled extreme precipitation found due to climate variability in the HiFLOR restored-SST experiments. An assessment using different climate models would therefore add value to allow for a sampling of risk across models, in addition to across experimental setups. These will be available shortly in the HighResMIP project (Haarsma et al 2016).

## 7.2 Future work and broader impacts

As described in the introduction and methods, we have purposefully focused our present assessment on one aspect of the flooding problem: the risk of extreme precipitation events that have the potential to produce inland flooding. We have provided provisional streamgauge data in the introduction (Figure 2) to illustrate the effect of the August 2016 event, but have not examined flood risks in the region from streamgauge data directly. Part of the reason for this is that real-time streamgauge data is provisional and subject to revision, which can be exacerbated during a flood when gauges can be overtopped and have missing data due to high water volumes or streamgauge malfunctions (Rantz 1982). The USGS advises users to cautiously consider the use of provisional streamgauge data for decision making (official USGS provisional policy available: <https://water.usgs.gov/wateralert/provisional/>). A complimentary modeling study of land surface conditions and interactions with the river environment also requires a more local modeling approach, potentially with a hydrologic model with information on the river system and small scale water processes, and conceivably including an estimate of the impact of direct human impacts (through urbanization, water diversion and management, etc.) which under our time constraints, data access, and present capabilities of our climate models was not feasible.





It is important to distinguish extreme precipitation events that are the topic of this study, motivated by the August
2016 rain event that led to devastating "freshwater" or "inland" flooding in south Louisiana, from events that lead to
"coastal" or "saltwater" flooding. In particular, the climate change context of saltwater flooding must include an assessment
of the regional sea level change contributions and meteorological conditions that can influence these types of events (e.g.,
Katsman et al, 2008, Sterl et al, 2012, Lin et al. 2012, 2014, Little et al. 2014). While certain meteorological conditions, such
as landfalling tropical cyclones, can lead to both freshwater and saltwater flooding (e.g., Lin et al. 2012, Villarini et al.
2014), the assessments and discussions presented here are only relevant to extreme rainfall events that have the potential to
initiate inland flooding; we do not address changes in storm surges, nuisance flooding (Moftakhari et al. 2015) or other
saltwater flooding events.
Dependence of the statics of extreme precipitation events in the Central U.S. Gulf Coast on large-scale climate
drivers could provide a scientific basis for seasonal predictions of the odds of these events, much as is now regularly done
for the statistics of hurricanes. However, as we show in Section 3.3, we are unable to find strong connections between the
statistics of these extreme precipitation events and modes of SST variability (e.g., ENSO), which suggests the possibility for
limited seasonal predictability for these events beyond the multi-decadal increase in probability from long-term climate
warming. However, potential sources of predictability may be uncovered by future refined analyses.
The extent to which the changing risk of extreme rainfall events like that in south Louisiana has implications for
stakeholders, such as homeowners, local and federal governments, the humanitarian system, and the insurance industry, will
depend on details of the exposure, vulnerability and the disaster preparedness and response strategies available to each.
Changes to the physical system are a key factor in  adaptation and decisions, but these factors operate in a complex
landscape. Through a disaster management lens, the increased frequency of this type of event found in this study may place
strains on humanitarian responders and institutions, especially in the future if this type of extreme event continues to become
more frequent. Knowing the change in return periods of the most extreme events can help to provide insight into how
humanitarian institutions can evolve to be prepared for the future; in addition to adapting to a broader trend of increasing
hydro-meteorological disasters globally (CRED 2015). A worthwhile topic to explore in further assessment of this and
related events is the extent to which public and media perception both before (local preparedness, willingness to evacuate)
and after (nationwide media coverage and awareness of impacts) may have been impacted by the fact that the storm was not
named. However, there is an insufficiency of peer-reviewed literature on this topic, even as media outlets in the UK and U.S.
have started naming winter storms following the German example (Cutlip 2013, Van Oldenborgh et al. 2015).
It is essential to note that this analysis has pursued an assessment of the climate context of extreme precipitation
events (a "climate attribution" study) in which we evaluate the impact of climate conditions and changes in radiative forcing
on the probability of extreme rainfall events in south Louisiana and the Central U.S. Gulf Coast. This analysis is
fundamentally different in nature from (and complementary to) assessments of the synoptic chain of events that led to the
particular Louisiana extreme precipitation event in August 2016 (we would label that "synoptic attribution"). Synoptic



attribution of the event generally involves a clear chain of events that led to the extreme rainfall event in a relatively deterministic fashion. Meanwhile, the climate attribution presented here is fundamentally probabilistic. Although we recognize that the synoptic context of this particular extreme event is unique (in fact all events are unique in detail), we have sought to understand the climate context of the probabilities of a class of events that causes extreme precipitation in the Central U.S. Gulf Coast of which this event (flood-inducing extreme precipitation in south Louisiana) is a member (Otto et al, 2016). Furthermore, it is possible to assess the climatic context in more detail, by assessing more proximate climate drivers than global-mean temperature or radiative forcing (e.g., by looking at the impact of particular patterns of SST), or by a more refined assessment of the detailed impact of the superposition of modes of climate variability and multi-decadal climate change (e.g., Delworth et al. 2015, Jia et al. 2016). For any particular event a spectrum of attribution studies (from purely synoptic to purely climate) could, and perhaps should, be pursued in order to unravel the various factors relevant to that event. Moreover, some of these studies are feasible at rapid attribution timescales while others require more time and focused resources to produce the specific and targeted modeling experiments and observational analyses.

Our ability to perform the climate attribution of this event was made possible by pre-existing multi-centennial global simulations with high spatial resolution models, which allowed us to efficiently assess the impact of radiative forcing changes on regional extreme precipitation events. These simulations, obviously, necessitated the long-term research aimed at developing these high-resolution models (e..g, Putnam and Lin 2007, Delworth et al. 2012, Vecchi et al. 2014, Murakami et al. 2015). Furthermore, this work was enabled a body of work using these models that provided the necessary understanding of the characteristics and fidelity of these models to simulate large-scale and regional climate, and weather events over a broad range of scales and phenomena (e.g., Vecchi et al. 2014; Msadek et al. 2014; Delworth et al. 2015; Jia et al. 2015, 2016; Murakami et al. 2015, 2016; Krishnamurthy et al. 2015; Zhang et al. 2015, 2016; Pascale et al. 2016; Van der Wiel et al. 2016).

In particular, this paper follows on a recent analysis of the climatology and $CO_2$ sensitivity of extreme precipitation events over the U.S. in these same models, showing that FLOR and HiFLOR in particular are uniquely capable of capturing Central U.S. Gulf Coast precipitation extremes, which has large biases in coarser resolution models (Van der Wiel et al. 2016). Though the analysis of extreme precipitation events in Van der Wiel et al. (2016) is of a different nature (focusing on much lower return period events, using different statistical methods, and focusing at the grid point scale rather than regional events), the results presented there are consistent with the current analysis. The previous paper showed that in response to increasing $CO_2$ levels in the atmosphere, precipitation extremes along the Central U.S. Gulf Coast increase in intensity, with less likely events exhibiting larger fractional intensity increases.

We have here sought to provide a scientifically rigorous rapid assessment of the climate context of this precipitation event, which had tragic consequences, to provide meaningful grounding to the public discussions of this event, given both the intense interest in this specific event and our ongoing work on the general subject of climate and extremes (and precipitation extremes in the U.S. in particular, van der Wiel et al. 2016). We hope that this study, including our explicit





discussion of the assumptions needed to pursue this accelerated assessment, will help push the scientific conversation
forward to improve our understanding of the risks and return periods of extreme precipitation in the Central U.S. Gulf Coast.
The field of rapid attribution analysis is still nascent and may one day lead to such assessments being the normal course of
action in response to an extreme event to help provide scientific basis for real-time discussions, and in longer-term disaster
response and rebuilding. Until that time, studies such as this will likely only be done for select regions and event types where
there is sufficient easily accessible data, and a team of scientists with the necessary expertise and ability to make time in their
schedules to provide a rapid assessment. We expect that these early efforts at event attribution will expand our knowledge
and capabilities on this subject, and facilitate further inquiry.

## Acknowledgements

We thank Geert Lenderink, Sarah Kew, Nathaniel Johnson, Kieran Bhatia and Fanrong Zeng for their helpful comments on
an earlier version of the manuscript. Funding for this work was supplied by the National Oceanic and Atmospheric
Administration, U.S. Department of Commerce to the Geophysical Fluid Dynamics Laboratory, to the Cooperative Institute
for Climate Science (award NA14OAR4320106). The statements, findings, conclusions, and recommendations are those of
the authors and do not necessarily reflect the views of the National Oceanic and Atmospheric Administration, or the U.S.
Department of Commerce, or other affiliated institutions. This project was made possible through generous support from
donors to Climate Central's World Weather Attribution initiative and the EU project EUCLEIA under Grant Agreement
607085. CPC U.S. Unified Precipitation data provided by the NOAA/OAR/ESRL PSD, Boulder, Colorado, U.S. and can be
downloaded from: from http://www.esrl.noaa.gov/psd/. USGS data was obtained from the automated website and are
provisional and subject to revision. The data are released on the condition that neither the USGS nor the United States
Government may be held liable for any damages resulting from its use.

## Data availability

NOAA GFDL climate model data is not readily available globally at all grid points and for all simulations owing to the size
of daily global climate model output for high resolution models with thousands of years of simulations (on the order of 100x
terabytes). We have made the precipitation data for the Central U.S. Gulf Coast, global temperature and ENSO data that
were used in this study available at the Climate Explorer: <http://climexp.knmi.nl/selectfield_att.cgi>.

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
