# Peer review of "Rapid attribution of the August 2016 flood-inducing extreme precipitation in south Louisiana to climate change"

_Hydrology and Earth System Sciences, 2016_

## Referee Comment (RC1) · Anonymous Referee #1 · 19 Oct 2016

Floods are one of the biggest natural disasters in the world – nearly 50% of damage all over the world is induced by river floods. Latest records show that last several decades from region to region frequency of large floods has changed. Heavy historical floods are damaging different regions and areas – a number of areas faced tremendous floods, induced by extremely heave rainfall, never observed before in the area. Changing flood frequency and its nature affect design flood parameters evaluation and thus have great interest for hydrological practice from region to region. Investigation of the climate change influence on such extreme events is of big interest and may describe further understanding of flood behavior. Thus presented topic holds rather high level of interest. Authors of the paper defined its objective as to perform rapid

attribution study of extreme precipitation, which caused devastating flooding event in south Louisiana. Technique they used incorporates long-term datasets observational point and gridded data analysis, two higher resolution global climate models, statistical analysis was based on well proven before Generalized Extreme Value Distribution, and made assumptions. Technique allows computation of return time change and increase in probability of the extreme event under present conditions and in reference interval (before significant anthropogenic load). Traditional techniques of design flood calculation may result in a significant error if they don't account for changing climate conditions. In the presented paper well tested technique is proposed of how one can estimate possible likelihood of extreme precipitation, and thus design flood characteristics change under the anthropogenic climate change influence in different regions (shown on south Louisiana case study). In the final part of the paper by providing the list of assumptions authors described limitations of the approach. In the result it can be said that authors fully met the given objective of the study. Results of the study are well presented with sufficient graphical ant table information, and described with good level of details, and thus provides comprehensive understanding of the presented material. Overall feeling about the paper is valuable and definitely recommended for reading of scientific community.

---

## Author Comment (AC1) · 20 Oct 2016

The authors would like to thank the reviewer for the positive comment regarding the submitted manuscript on the Louisiana precipitation event.

---

## Referee Comment (RC2) · Anonymous Referee #2 · 20 Nov 2016

The study aims at attributing the rainfall leading to widespread flooding in Louisiana in August 2016 to anthropogenic climate change. At least according to the title of the study. The manuscript provides a thorough analysis of the extreme rainfall event and includes an assessment of the role of anthropogenic climate change as well as El Nino. The attribution part of the presented manuscript could be stronger and the group has presented better studies in terms of robustly attributing the role of anthropogenic climate change as this study is primarily based of one model and focused more on a general climatological context than the anthropogenic signal per se. Analyses of the British rainstorm and the French rainfall extremes submitted to the same journal by a similar set of authors better harness the power of multiple methodologies and multi

models. I realise I disagree with some vocal colleagues on this aspect thus it's worth highlighting. Adding for example an assessment based on the very large ensembles from the Oxford group which allow for a clear separation of the anthropogenic climate signal as in previous studies could have made the attribution statement stronger (even if the statement would have been that the model doesn't capture the events at all but judging from their website a regional model in the right part of the world seems to exist).

The presented studies is however a very strong assessment of the rainfall leading to the Louisiana floods in terms of the nature of the extreme event, the atmospheric processes and the rareness of the event in todays climate making it a valuable contribution to the literature on such extremes. In particular the clearly presented model evaluation and a presentation of the assumptions that went into the conclusions could serve as a model how these kinds of analyses could and maybe should be done in the future. I would thus recommend only relatively minor revisions to make the presentation of results clearer but would advise on changing the title to make it clear that the attribution to man-made climate change it not the main focus but rather and autopsy of the event in a climate context.

Furthermore, currently the discussion section is rather lengthy and while of very high quality in the first part when describing the assumptions made in the meteorological analysis and their discussion the second part (7.2) relating to the impacts of such an event does not provide any original analysis but rather common place arguments that apply in any extreme event context. This stark decrease in the quality of the discussion does not serve the overall paper well and leaves the reader unnecessarily disappointed when reading the end of a very good paper. I would thus recommend drastically shortening section 7.2 and simply stating that in the future an analysis of the specific vulnerabilities and exposure in the impacted region would enhance the direct applicability of the meteorological event autopsy. The latter part of the section is better placed in the methodology section but has already been mentioned there.

Specific comments

p.2, l.35 – strange grammar, rephrase sentence

p.2, l.52 – strange grammar, rephrase

p.3, l.72- explain "flood stage"

p.4, caption figure 2 – what is the pink area?

p.4, ll.86-92 – while not irrelevant, the introduction is already very long, to actually have someone read this interesting paper it would be a good idea to shorten the introduction, this paragraph provides a good opportunity, but not the only one

p.5, l.124 – add reference to what to examples or more explanation of what is meant by thermodynamic and dynamic responses and 'weather type'. In particular the latter is quite ambiguous

p.6, ll.143/144 – to avoid even more confusion than already exists in the community the term "event attribution" with or without added "extreme" or "probabilistic" would be better

p.7, ll.181-184 – what are the criteria to through out the other two considered models?

p.7., l.198 & above – flux adjustment is not done in many models anymore for good reasons, very briefly discuss what the implications are for your study

p.9, l.241 – mention that in 7.1 you'll find the assumption not unjustified

p.10, l.266 – add "a" between has & realistic

p.10, l.270 – do you mean the Golf region?

p.11, l.295 – add reference for the moving block technique

p.13,l.339 – why do you use 3-day averages instead of sums?

p.13,ll.352-355 – is there a reason for expressing the change in probability by a factor and the intensity in percentages? If not use the same measure for both.

p.14, l.361 – In which of the 2 sentences does figure 1 d,f belong?

p.24,l.542 – up to 2100

p.25,ll.552-553 – this is a much better expression of your bias correction method than before

p.26,l.580 – is this a justified assumption?

p.28, table 3 – make clear in the table which experiment provides the proper attribution analysis

p.29, figure 15 – nice figure, but instead of using colours associated with models and in effect giving the same information twice it would be nice for the reader to have the proper attribution analysis highlighted instead

p.30, l.537 – management of what?

p.30, section 7.1 – are the assumptions in any particular order? If so, what is it, if not it might be worth saying that.

p.30, assumption 1) While you discuss later whether the grouping is justified or at least how it can be tested you do not comment on the GEV fit at all. Is that justified? Can we test that?

p.32, assumption 9) – this is the strongest assumption I think and one that could be tested by using e.g. the weather@home model you've mentioned before, might be worth discussing why that has not been used (I assume no suitable runs where available but given that it is part of all other studies of rapid attribution publicised under World Weather Attribution for good reason it is worth mentioning)

p.32, l.720 – Finally? It is not your final point here. . ..

p.33, ll.737-748 – this is a paragraph that could be easily written in a single sentence

p.33, l.751 – why did you not include the satellite data in the study? I'd assume it

shouldn't take longer to analyse than the stations.

p.34,l.761 – for the kind of analysis you do high resolution seems only one factor, ensemble size is the other

section 7.2 – this section is rather disappointing in quality & added value given that it contains no insights in vulnerability and the latter part reads like an acknowledgement. I would recommend rewriting and drastically shortening to say why your study adds value instead of listing at length all the things you haven't done. It would be sufficient to mention that impact modelling and vulnerability assessment would be a great complementation.

p.35,ll.3303 – end – it does not become clear how a synoptic analysis would provide anything useful in terms of decision making unless it is coupled with an assessment of the likelihood of the synoptic systems to occur hence this paragraph becomes a bit incomprehensible, again it would be better to highlight the strengths of your study and maybe refer to the Otto et al. paper if you want to highlight that the topic is not uncontroversial

---

## Author Comment (AC2) · 9 Dec 2016

*Hydrol. Earth Syst. Sci. Discuss.,*
*doi:10.5194/hess-2016-448, 2016*

**Reply to the interactive comment of Referee #2**

The authors would like to thank the reviewer for the detailed commentary on the submitted manuscript. Below we provide replies to the suggestions that were made. It is our hope that these changes have improved the clarity of the presented results.

The authors agree that the final section of the paper is very lengthy. As the reviewer points out, the list of crucial assumptions provides a careful frame for the current study and we hope it will guide future studies of the Louisiana flooding event. In Section 7.2 we widen our view, and discuss further work regarding the flooding event and synoptic situation. We are of the opinion that it is important to mention these open questions, and to point out that probability attribution is not the only relevant study for this tragic event.

The authors disagree with the reviewer that the main focus of the paper is 'an autopsy of the event in a climate context'. Indeed we report on the average return times of comparable precipitation events in the present climate, however the computation of these numbers are necessary steps towards calculating changes in statistics and providing the attribution to anthropogenic climate change.

As noted, including data from multiple models could strengthen the attribution statement. Unfortunately the models used for the attribution of the French and German precipitation extremes (HadGEM3-A N216 and EC-Earth 2.3 T159, Van Oldenborgh et al., 2016) were found to be unable to realistically simulate the full distribution of precipitation events on the Central U.S. Gulf Coast. Including data from these models would therefore not have strengthened the conclusions, as the only relevant statement that could be made is that these models do not capture the extremes and are therefore unsuitable for the analysis. The ensemble of regional model experiments from Weather@Home for the region of interest (Central America, CAM50, CAM25) is still in progress, for future analysis these experiments might indeed add valuable data when they become available.

***Specific comments***

p.2, l.35 – strange grammar, rephrase sentence

The sentence will be modified.

> *… The global climate models tell a similar story,* **in the most accurate analyses the regional probability of 3-day extreme precipitation increases by more than a factor 1.4 due to anthropogenic climate change.** *…*

p.2, l.52 – strange grammar, rephrase

The sentence will be modified.

> *… At that time the National Hurricane Center stated that* **the low pressure system** *might transform into a tropical depression* **if it moved to the** *Gulf of Mexico (Schleifstein, 2016). …*

p.3, l.72- explain "flood stage"

Flood stage is generally defined as the gage height at which overflow of the natural banks of a stream or river starts causing damage in the area. We will add this description to the manuscript.

> *… (USGS) registering above flood stage levels* **(levels at which overflow of natural banks starts to cause damage in the local area)** *at 30 sites and…*

p.4, caption figure 2 – what is the pink area?

The shaded pink area indicates the 3-day period of maximum precipitation over the Central U.S. Gulf Coast (12-14 August). The figure caption will be modified to include this information.

> *… on the Amite River.* **Shaded pink areas indicate the 3-day period of maximum precipitation (12-14 August 2016).** *…*

p.4, ll.86-92 – while not irrelevant, the introduction is already very long, to actually have someone read this interesting paper it would be a good idea to shorten the introduction, this paragraph provides a good opportunity, but not the only one

The authors believe it is important to not the large societal impact of the event. This study was in part motivated by these significant human impacts of the Louisiana flooding event, as also noted in the introduction. We will shorten the paragraph slightly.

p.5, l.124 – add reference to what to examples or more explanation of what is meant by thermodynamic and dynamic responses and 'weather type'. In particular the latter is quite ambiguous

Weather types refer to the previously described weather systems and events that may cause precipitation extremes. To avoid confusion and for consistency, we will replace the phrase 'weather type' with 'weather system'.

We will add references to a review paper that describes different thermodynamic and dynamic responses. To avoid adding additional length to the introduction (see previous point) we won't give an extensive list.

> *… may be different for* **different** *weather* **systems (O'Gorman 2015)***.*

p.6, ll.143/144 – to avoid even more confusion than already exists in the community the term "event attribution" with or without added "extreme" or "probabilistic" would be better

We will change the term to 'event attribution'.

> *… upon these methodologies for* **event attribution** *and also explores …*

p.7, ll.181-184 – what are the criteria to through out the other two considered models?

As in the French & German precipitation extremes study, we fitted the extreme tail of 3-day precipitation in the study area (a single representative grid point at the time) and compared the fit parameters of the GEV with those of the observations (copied straight from the lab book, not truncated to sensible precision):

| (in 2016) | $\mu'$ | $\sigma'$ | $\xi$ | $\alpha$ |
|---|---|---|---|---|
| obs | 51.792 (50.866... 52.786) | 13.798 (13.137... 14.231) | 0.118 (0.077... 0.136) | 10.735 (7.701... 10.485) |
| HadGEM3-A | 35.141 (34.199... 36.223) | 12.982 (11.940... 13.576) | 0.156 (0.107... 0.215) | 1.539 (-1.779... 6.275) |
| EC-Earth 2.3 | 26.896 (26.603... 27.119) | 5.165 (4.939... 5.340) | -0.034 (-0.072... -0.008) | 1.502 (0.884... 2.135) |

HadGEM3-A N219 has a 30% lower location parameter but almost the same scale parameter, hence much more variability in the high extremes when bias-corrected to the correct mean. EC-Earth T159 only has 50% of the location parameter, and not enough variability even after scaling with the location parameter and a wrong (negative) shape parameter. We concluded that neither model can represent the statistical properties of extreme precipitation in the study area.

The physical consideration is that the ~150 km resolution of EC-Earth T159 is not high enough to represent tropical storms well enough to capture their precipitation characteristics. The experience with FLOR vs HiFLOR in this area (Van der Wiel et al, 2016) suggests that the 60km resolution of HadGEM3-A N219 is also not sufficient.

p.7., l.198 & above – flux adjustment is not done in many models anymore for good reasons, very briefly discuss what the implications are for your study

We will add a comment on the implications.

> *… observed climatological state.* **This procedure reduces model biases of for example SSTs, tropical cyclones (Vecchi et al. 2014) and precipitation patterns. We assume the modeled response to changes in radiative forcing are not impacted by the flux-adjustment (see Section 7.1).** *The adjustment method is …*

p.9, l.241 – mention that in 7.1 you'll find the assumption not unjustified

We will add a comment.

> *and assume that the impact of the slow climate drift in each model experiment on the statistics of precipitation extremes is small* **(see justification in Section 7.1)**.

p.10, l.266 – add "a" between has & realistic

We will modify the sentence.

> *… 29-31 ºN, 85-95 ºW, which has* **a** *relatively homogenous average precipitation extreme magnitude …*

p.10, l.270 – do you mean the Golf region?

Yes indeed, we will add the specification.

> *For some analyses we then take the maximum over the* **Central U.S. Gulf Coast** *region.*

p.11, l.295 – add reference for the moving block technique

We will add the reference.

> *Efron, B. and R.J. Tibshirani, R. J., 1998. An introduction to the bootstrap, Chapman and Hall, New York. 439pp.*

p.13,l.339 – why do you use 3-day averages instead of sums?

For an attribution study to be completed within a rapid timeframe, analysis methods must be set up and ready to go before the event. In the current set up, the analysis tools (KNMI Climate Explorer) give average precipitation. Whether averages or sums are noted does not impact the conclusions, the average results may simply be multiplied by a factor 3 to get sums.

p.13,ll.352-355 – is there a reason for expressing the change in probability by a factor and the intensity in percentages? If not use the same measure for both.

The reason is that the changes in probability are usually much larger (one easily gets a factor four) than chances in intensity (tens of percents). Also, the two are often confused in popular accounts, keeping the units separate minimises the risk of this confusion.

p.14, l.361 – In which of the 2 sentences does figure 1 d,f belong?

There should be a reference to Figure 3d,f here, it will be corrected. It is then also clear that the reference is relevant for the sentence before.

> … of 17% (C.I. 10%-21%), **Figure 3d,f**. The increase in probability …

p.24,l.542 – up to 2100

We will modify the text.

> does not change in the model world **up to** 2100, in spite of a different mix p.25,ll.552-553 – this is a much better expression of your bias correction method than before

We will use this expression in the previous case (FLOR-FA, line 536 of the revised manuscript) and refer to that explanation in this place.

> Because of model bias, we d**efine our event to have the same return period as the gridded observations in 2016 (**around 30 years**, 115 mm/day).**

> We correct for this bias **as we did for the FLOR-FA experiment (the 30 year event is** 103 mm/day).

p.26,l.580 – is this a justified assumption?

Here, we assume forced changes over the period 1971-2015 dominates over internal changes, for example due to ENSO. ENSO is high-frequency and cancels out over this fairly long period: there have been many El Niño and La Niña events. Considering longer time scale modes of climate variability, the PDO also has trend very close to zero over this time period. The AMO has a strong trend, hence our investigation whether extreme rainfall in this area is connected to low-frequency Atlantic variability. The connection we found (with very low field significance, so it may have been spurious) would have lowered the probability of high precipitation, i.e., counter-acted the global warming signal.

p.28, table 3 – make clear in the table which experiment provides the proper attribution analysis

All model experiments mentioned in table 3 are used to create the attribution analysis. We will add a note making this explicit.

> …Gulf Coast. **Note the modeled changes can be attributed to anthropogenic climate change.**

p.29, figure 15 – nice figure, but instead of using colours associated with models and in effect giving the same information twice it would be nice for the reader to have the proper attribution analysis highlighted instead

As described at the previous point, all model experiment data contribute to the attribution. The orange/red colours thus indicate the values that have be used for attribution. Values based on observational data are on purpose coloured very differently (blue), to make the suggested distinction.

p.30, l.537 – management of what?

The sentence is indeed unclear. We will remove the statement on management and rather just mention societal impacts. The management implications (policy making etc.) may be considered to be part of societal impacts.

> *… Finally, we note some societal impacts of the findings.*

p.30, section 7.1 – are the assumptions in any particular order? If so, what is it, if not it might be worth saying that.

The assumptions aren't in a particular order, though they are somewhat grouped by similarity. At the start of section 7.1 there is a list of topics for the crucial assumptions, we will modify the order to represent the order of the actual list.

> *… we have made the following crucial assumptions about* **the statistical distribution of precipitation extremes, the observations, the relationship between temperature and precipitation extremes and the models***. We have tested …*

p.30, assumption 1) While you discuss later whether the grouping is justified or at least how it can be tested you do not comment on the GEV fit at all. Is that justified? Can we test that?

We write that the appropriateness of GEV fits should be tested (line 723 of the revised manuscript). A possible way to test this would be to analyze very long model integrations or very large ensembles, for which one can accurately compute the 1-in-500-year event and for which one can compute GEV statistics based on shorter blocks of modelled data.

p.32, assumption 9) – this is the strongest assumption I think and one that could be tested by using e.g. the weather@home model you've mentioned before, might be worth discussing why that has not been used (I assume no suitable runs where available but given that it is part of all other studies of rapid attribution publicised under World Weather Attribution for good reason it is worth mentioning)

The regional Weather@Home ensemble experiment for Central America (which includes the region of interest) is in progress and is not available at present.

p.32, l.720 – Finally? It is not your final point here. . ..

We will replace finally with furthermore.

> *… and Sutton 2009).* **Furthermore***, the appropriateness of GEV fits in general.*

p.33, ll.737-748 – this is a paragraph that could be easily written in a single sentence

The authors are of the opinion it is important to discuss the current scientific understanding on thermodynamic and dynamic changes separately and discuss how the former likely does not impact our results, while the latter might impact the presented changes. This is an active area of research that we wish to recognize in full.

p.33, l.751 – why did you not include the satellite data in the study? I'd assume it shouldn't take longer to analyse than the stations.

The period covered by satellite data (roughly 1979-present) is much shorter than the period covered by station observations (for this region 1891-present). The GEV estimates of return periods are better and more reliable if more data is included, therefore we have chosen to focus on two long datasets of observed station data. Secondly, satellite estimates of precipitation do not correspond well to ground-based observations, with biases that vary over time, making them ill-suited for trend analyses.

p.34,l.761 – for the kind of analysis you do high resolution seems only one factor, ensemble size is the other

High resolution is important for these model experiments, because tropical cyclones (and pseudo-tropical cyclones like the one that occurred in August 2016) need to be adequately simulated; the characteristics of tropical cyclones are more poorly simulated in models with resolutions of ~200km that are typical of the CMIP5 ensemble, but the two models used here have relatively realistic simulation of the statistics and characteristics (including rainfall distribution) of tropical cyclones (e.g., Vecchi et al. 2014; Murakami et al. 2015, 2016; Liu et al. 2016). Furthermore, in order to assess the statistics of the return times we require relatively long simulations (many 100s of years total), and we are unaware of any other high-resolution models that have been run for that length of time.

An ensemble of models with comparable capability to simulate the phenomena that lead to extreme rainfall in this region, and with experiments of sufficient length to assess the statistics, would allow one to further explore the structural uncertainty, but such an ensemble does not yet exist. We plan to repeat the analysis when these data have become available in HiResMIP.

section 7.2 – this section is rather disappointing in quality & added value given that it contains no insights in vulnerability and the latter part reads like an acknowledgement. I would recommend rewriting and drastically shortening to say why your study adds value instead of listing at length all the things you haven't done. It would be sufficient to mention that impact modelling and vulnerability assessment would be a great complementation.

In Section 7.1 we list and work through all assumptions and further work directly related to the present study. Section 7.2 is used to widen our vision, that was narrowed to only include extreme precipitation events after the introduction, and discuss avenues of further scientific investigation related to the flooding event. In our opinion, because of the rapid timeframe of the current study and its publication being one of the first on the Louisiana flood-inducing event, it is our task to explicitly mention that the current work is not the complete story.

The last three paragraphs (line 824 and onwards) provide insight into the procedure of rapid assessment. Though attribution analysis has become more routine since its introduction in 2005, the rapid attribution is relatively new. It is our hope that among the broader impacts of this study it serves the scientific community with a tool of how we believe such a study must be done. Therefore, we describe the modelling efforts that were already in place, without which the analysis could not have been done. And we describe a recent verification study of extreme precipitation in these models and in the region of interest, which was crucial in the decision to qualify these models for the study.

To address the confusion noted by the reviewer, we will alter the text.

> *Climate attribution studies such as this one can only be performed with* pre-existing multi-centennial global simulations with high spatial resolution models*. This* allowed us to efficiently assess the impact of radiative forcing changes on regional extreme precipitation events. …

p.35,ll.3303 – end – it does not become clear how a synoptic analysis would provide anything useful in terms of decision making unless it is coupled with an assessment of the likelihood of the synoptic systems to occur hence this paragraph becomes a bit incomprehensible, again it would be better to highlight the strengths of your study and maybe refer to the Otto et al. paper if you want to highlight that the topic is not uncontroversial.

We describe how a synoptic attribution is fundamentally different from an probability attribution presented in the study. Each study is scientifically relevant, and each serves society in a different manner. Synoptic studies serve weather and emergency services, probability studies are more relevant for risk management in long-term policy. Otto et al. 2016 is presently provided as a reference to direct the reader to an exemplary paper on the topic for further reading.

[revised manuscript text omitted]

---

## Author Response (AR1)

*Hydrol. Earth Syst. Sci. Discuss.,*
*doi:10.5194/hess-2016-448, 2016*

**Reply to the comments of the Editor**

Editor Decision: Publish subject to minor revisions

Comments to the Author:

The Referees were very positive about your publication and requested only minor revisions.

I have one additional minor request which is to highlight/summarise the main assumptions made in the method in the beginning paragraphs of the paper (which are provided in detail in later in the paper) and also highlighting which parts of dynamic earth system connectivity you are and are not testing, so it is very clear to the reader what rapid attribution can and cannot provide. This no doubt will be a very well read paper.

The authors would like to thank the editor for her time and efforts. In response to the request, we have added a short description of the crucial assumptions and point the reader to Section 7, which provides the full discussion. Furthermore, we have added a few sentences of the limitations of the study.

[revised manuscript text omitted]